# On Learning Parallel Pancakes with Mostly Uniform Weights

Ilias Diakonikolas [1]   Daniel M. Kane [2]   Sushrut Karmalkar [3 4]   Jasper C.H. Lee [5]   Thanasis Pittas [1]

## Abstract

We study the complexity of learning $k$-mixtures of Gaussians ($k$-GMMs) on $\mathbb{R}^d$. This task is known to have complexity $d^{\Omega(k)}$ in full generality. To circumvent this exponential lower bound on the number of components, research has focused on learning families of GMMs satisfying additional structural properties. A natural assumption posits that the component weights are not exponentially small and that the components have the same unknown covariance. Recent work gave a $d^{O(\log(1/w_{\min}))}$-time algorithm for this class of GMMs, where $w_{\min}$ is the minimum weight. Our first main result is a Statistical Query (SQ) lower bound showing that this quasi-polynomial upper bound is essentially best possible, even for the special case of uniform weights. Specifically, we show that it is SQ-hard to distinguish between such a mixture and the standard Gaussian. We further explore how the distribution of weights affects the complexity of this task. Our second main result is a quasi-polynomial upper bound for the aforementioned testing task when most of the weights are uniform while a small fraction of the weights are potentially arbitrary.

## 1. Introduction

Learning mixture models in high dimensions is a classic and fundamental task with applications in a plethora of domains, such as bioinformatics, astrophysics, and marketing (Lindsay, 1995; García-Escudero et al., 2010); see Titterington et al. (1985) for an extensive list of applications. The prototypical case is that of Gaussian Mixture Models (GMMs)

which is one of the most studied problems in statistics and machine learning, with a large body of research over the past few decades, e.g., Vempala & Wang (2002); Kannan et al. (2005); Achlioptas & McSherry (2005) — see Appendix A for a detailed literature review.

The setup is as follows: the learning algorithm observes i.i.d. samples from a $k$-component GMM model ($k$-GMM) in $\mathbb{R}^d$, $P = \sum_{i=1}^{k} w_i \mathcal{N}(\mu_i, \Sigma_i)$, and the goal is to either learn the mixture in total variation distance, learn its parameters, or cluster samples from the GMM correctly. The first task is known to be information-theoretically feasible with $\mathrm{poly}(d, k)$ samples, as are the second and third, provided the components are sufficiently well-separated, however known algorithms often require more. While the particular case of spherical mixtures (i.e., $\Sigma_i = I$) can be learned in $\mathrm{poly}(d, k)$ time and samples, (Liu & Li, 2022; Diakonikolas & Kane, 2024), the best-known algorithms for learning arbitrary GMMs (i.e. with arbitrary weights, and arbitrary and different component covariances) have sample complexity that scales with $d^{O(k)}$ (Bakshi et al., 2022). In this paper, we are concerned with an intermediate regime between the two extremes, where the components share an unknown but common covariance matrix.

Diakonikolas et al. (2017) showed that for such mixtures, any sub-exponential time algorithm in the *Statistical Query* (SQ) model requires a sample complexity of $d^{\Omega(k)}$. The SQ model consists of algorithms that, instead of drawing samples from the data distribution, make queries to approximate expectations of bounded functions (formally defined in Definition 1.2).

The hard instances they proposed are "parallel pancakes" GMMs—mixtures of pairwise-separated Gaussians whose component means are collinear along an unknown direction $v$, with arbitrary variance in the $v$-direction and identity covariance in the orthogonal subspace. This will be formally defined in Problem 1.1. Bruna et al. (2021); Gupte et al. (2022) further extended the hardness result to general algorithms but under cryptographic assumptions; similar hardness results were also shown for sum-of-squares algorithms (Diakonikolas et al., 2024).

While these results together might suggest that $k$-GMM learning is fully understood algorithmically, the current theory remains unsatisfactory, in the following sense: the

Author names in alphabetical order [1]Department of Computer Sciences, University of Wisconsin Madison, Madison, United States [2]University of California San Diego, San Diego, United States [3]Microsoft Research, Cambridge, England [4]Some of this work was done while the author was a postdoctoral researcher at UW-Madison. [5]University of California Davis, Davis, United States. Correspondence to: Thanasis Pittas <pittas@wisc.edu>.

*Proceedings of the 42$^{nd}$ International Conference on Machine Learning*, Vancouver, Canada. PMLR 267, 2025. Copyright 2025 by the author(s).

hard instances developed in Diakonikolas et al. (2017) have rather ill-conditioned mixing weights—some of the mixing weights are $1/\operatorname{poly}(k)$, but others can be as small as $2^{-k}$. A natural question then is: is it possible to improve the complexity of learning algorithms when all weights are more naturally conditioned, i.e., $w_i \geq 1/\operatorname{poly}(k)$ for all $i$?

This question was considered in Buhai & Steurer (2023); Anderson et al. (2024), which study GMMs that have a minimum mixing weight $w_{\min} \geq 1/\operatorname{poly}(k)$ and unknown but common covariance across components. Under the assumption that the mixture components are separated in total variation distance, they provide an algorithm that can correctly cluster 99% of the points, using time and sample complexity $d^{\log(1/w_{\min})} \leq d^{O(\log k)}$. In particular, their results apply to parallel pancake instances, showing that it is possible to circumvent the $d^{\Omega(k)}$ (SQ) lower bound under mixing weight assumptions.

These prior results on learning mixtures with restricted mixing weights serve as motivation and the starting point of the present work. In particular, the first question we study is:

> *Is it possible to substantially improve the algorithm of Anderson et al. (2024) to a $\operatorname{poly}(d, k)$ time algorithm for parallel pancakes when each $w_i \geq 1/\operatorname{poly}(k)$?*

Our first main result rules out this possibility for SQ algorithms. Specifically, we show in Theorem 1.3 that even when the mixing weights are uniform, any SQ algorithm for such instances requires $d^{\Omega(\log k)}$ complexity. In fact, the lower bound holds even for the more basic task of distinguishing between a $k$-GMM from that family and $\mathcal{N}(0, I)$.

Our second question stems from the fact that the algorithm in Anderson et al. (2024) has complexity $d^{\log(1/w_{\min})}$, meaning that a single point with arbitrarily small weight (e.g., $2^{-k}$) can result in $d^k$ complexity.

> *What is the correct complexity dependence on $w_{\min}$ for learning $k$-component parallel pancakes?*

Specifically, we consider again the testing problem of distinguishing between a $k$-parallel-pancake GMM and $\mathcal{N}(0, I)$, but where $k' \leq k$ components can have arbitrary weights while the remaining $k - k'$ points must have uniform weights. We show that this mixing weight restriction implies that the testing problem can be solved with time and sample $(kd)^{O(k'+\log k)} + (\log k)/w_{\min}$ — an inverse-linear dependence on $w_{\min}$ instead of quasi-polynomial as suggested by the Anderson et al. (2024) result. While this testing upper bound does not imply a general learning algorithm $k$-GMM, it serves as a first step in understanding the nuances of the computational landscape of GMMs with respect to the assumptions on the mixing weights.

The technical core for both our main results is to deter-

mine the maximum number $m$ of moments that $k$-parallel-pancake GMMs can match with $\mathcal{N}(0, I)$. Our SQ lower bound (Theorem 1.3) comes from showing that $m = \Omega(\log k)$, by employing a result from design theory. Our second, algorithmic result (Theorem 1.4) critically builds on an impossibility-of-moment-matching argument (Proposition 4.1), showing that if there are $k' \leq k$ arbitrary weights in the $k$-GMM, then $m$ must be $O(\log(k) + k')$. We show this through a novel proof strategy that bounds the ratio of expectations of appropriately chosen non-negative polynomials that vanish on the points with arbitrary weights.

### 1.1. Our Results

We first formally state the hypothesis testing problem which requires the algorithm to distinguish between a $k$-parallel pancake and the standard Gaussian $\mathcal{N}(0, I)$.

**Problem 1.1** (Parallel Pancakes Testing Problem)**.** One has (i.i.d. sample or SQ) access to a distribution $D$ where either:

- (Null Hypothesis) $D = \mathcal{N}(0, I)$.
- (Alternative Hypothesis) $D$ is a Gaussian mixture of the form $\sum_{i \in [k]} w_i \mathcal{N}(v\mu_i, I - \delta vv^\top)$, for some unit vector $v \in \mathcal{S}^{d-1}$, centers $\mu_i \in \mathbb{R}$, and weights $w_i \geq 0$ for $i \in [k]$, with $\sum_{i \in [k]} w_i = 1$. That is, $D$ is a $k$-GMM with collinear centers and variance $1 - \delta$ along the direction of the centers and $1$ in every orthogonal direction.

The goal is to distinguish between the two cases.

Before presenting our first main result, we recall the definition of SQ algorithms. These algorithms, instead of directly accessing samples, query expectations of bounded functions of the distribution. The SQ model, introduced in (Kearns, 1998), has since been extensively studied in various contexts (Feldman, 2016). Many supervised learning algorithms, and several known machine learning techniques are implementable using SQs (Feldman et al., 2017a;b).

**Definition 1.2** (STAT Oracle)**.** Let $D$ be a distribution on $\mathbb{R}^d$. A statistical query is a bounded function $f : \mathbb{R}^d \to [-1, 1]$. Given $f$ and an accuracy parameter $\tau > 0$, $\operatorname{STAT}(\tau)$ returns a $v \in \mathbb{R}$ such that $|v - \mathbf{E}_{x \sim D}[f(x)]| \leq \tau$.

Since a call to $\operatorname{STAT}(\tau)$ can be simulated in the standard PAC model by averaging $1/\tau^2$ samples, $\tau$ serves as the SQ model's analog to sample complexity. An *information-computation* tradeoff in the SQ model states that any SQ algorithm for a given problem must either make a large number of queries or at least one query with very fine accuracy (which informally implies a tradeoff between sample complexity and runtime in the standard PAC model). We are now ready to state our first main result.

**Theorem 1.3** (SQ Lower Bound for Uniform Weights)**.** *Let $C$ be a sufficiently large absolute constant, $k > C$ and $d \geq (\log k \log d)^2$ be integers. If we further restrict the*

*alternative hypothesis in Problem 1.1 to have $w_i = 1/k$ for all $i \in [k]$, any SQ algorithm requires either $2^{d^{\Omega(1)}}$ queries or at least one query of accuracy $d^{-\Omega(\log k)}$.*

**Remarks** Buhai & Steurer (2023); Anderson et al. (2024) presented an algorithm for solving Problem 1.1 using $d^{O(\log k)}$ time and samples (e.g., Theorem 1.1 in the first paper, which was the first to achieve this). Our Theorem 1.3 shows that this complexity is best possible. Notably their work requires the components to be statistically separated, but this is something that we can also ensure by taking $\delta$ sufficiently small (since $\delta$ does not affect the complexity lower bound).

We now move to our second main result.

**Theorem 1.4** (Testing Algorithm for Parallel Pancakes). *Consider the version of the parallel pancakes hypothesis testing problem (Problem 1.1), where $k' \leq k$ of the weights $w_i$ in the Gaussian mixture are unconstrained and the remaining $k - k'$ are assumed to be equal to each other. There is an algorithm for that problem which draws $n = O\left((kd/\delta)^{O(k'+\log(k))} + \log(k)/w_{\min}\right)$ samples (where $\delta$ is as in Problem 1.1 and $w_{\min} = \min_{i \in [k]} w_i$ is the smallest weight), has runtime polynomial in $n, d$, and it outputs the correct hypothesis with probability at least $0.99$.*

The algorithm is based on estimating the first $O(k' + \log k)$ moment tensors through the empirical tensors, and thus it is also naturally expressible in the SQ model.

**Remarks** A single component with arbitrarily small weight can make the complexity in (Buhai & Steurer, 2023; Anderson et al., 2024) blow up quasipolynomially. By contrast, our algorithm can handle any number of such points, and the complexity interpolates smoothly between the all-uniform and the fully general weights cases.

### 1.2. Overview of Techniques

For Theorem 1.3, it suffices to show existence of a one-dimensional distribution (corresponding to the projection along the hidden direction $v$ in the parallel pancakes mixture in Problem 1.1) that matches a lot of moments with $\mathcal{N}(0,1)$ and is thus hard to distinguish. Concretely, the goal is to show the existence of a set $S \subset \mathbb{R}$ of size $k$ such that $\mathbf{E}_{x\sim S}[x^i] = \mathbf{E}_{x\sim\mathcal{N}(0,1)}[x^i]$ for all $i = 1, \ldots, t$, where $t = \Omega(\log k)$ and $x \sim S$ denotes the uniform distribution on $S$. Once established, the theorem follows from standard SQ theory: convolving this discrete distribution with a narrow Gaussian yields a $k$-GMM $B$ that still matches the first $t$ moments with $\mathcal{N}(0,1)$. A standard result from (Diakonikolas et al., 2023) then shows that hiding $B$ along an unknown direction is hard to distinguish from $\mathcal{N}(0,I)$.

Fortunately, the desired moment-matching construction,

known as a $t$-design, has been well-studied. Kane (2015) shows that designs of small size to match the moments of a distribution $Q$ exist when the support of $Q$ is "path-connected". The design's size is upper bounded by the number $K$, which is defined to be the supremum of the ratio $\frac{\sup_{x\in X} p(x)}{|\inf_{x\in X} p(x)|}$ taken over all degree-$t$ zero-mean polynomials $p$. Thus, it suffices to show $K = 2^{O(t)}$ to prove Theorem 1.3. However, since the Gaussian distribution has unbounded support, there are (many) polynomials $p$ where $\sup_{x\in\mathbb{R}} p(x)$ is infinite while the infimum is clearly finite.

To address this, we can instead consider another distribution $Q$ supported on an interval $I$ of length $O(\sqrt{t})$, which also matches the first $t$ moments with $\mathcal{N}(0,1)$ (Lemma 3.3 from Diakonikolas et al. (2017)). Thus, by creating a design to match $Q$ we only need to bound $K$ with $X = I$, which is now possible (Lemma 3.5). Specifically, by expressing $p$ in the Hermite basis, we can show that $\sup_{x\in I} p(x) \leq \mathbf{E}_{x\sim\mathcal{N}(0,1)}[|p(x)|]2^{O(t)}$. Additionally, by Gaussian anti-concentration, we can show that any zero-mean polynomial $p$ of degree $t$ has a $2^{-O(t)}$ probability of being less than $-2^{-O(t)}\mathbf{E}_{x\sim\mathcal{N}(0,1)}[|p(x)|]$. This shows that $|\inf_{x\in I} p(x)| \geq \mathbf{E}_{x\sim\mathcal{N}(0,1)}[|p(x)|]2^{-O(t)}$.

We can also show that this bound is tight: if $A$ is the uniform distribution on $k$ points, it cannot match more than $O(\log k)$ moments with $\mathcal{N}(0,1)$. The argument is that for any non-negative function $f$, $\frac{\mathbf{E}_{x\sim A}[f^2(x)]}{\mathbf{E}_{x\sim A}[f(x)]^2} \leq k$. If $A$ matches the first $4t$ moments with $\mathcal{N}(0,1)$, setting $f(x) = x^{2t}$ makes the ratio $2^{\Omega(t)}$, implying $t = O(\log k)$.

In fact, we can extend the result to non-uniform distributions where all but $k'$ points in the support have equal weight, showing that such distributions cannot match more than $O(\log(k)+k')$ moments with $\mathcal{N}(0,1)$ (Proposition 4.1). As we will explain later, this will lead to the testing algorithm in Theorem 1.4.

The first step towards Proposition 4.1 is to show that, if all but $k'$ points have weight at least $w_0$, then it is impossible to match more than $O(\log(1/w_0)+k')$ moments with $\mathcal{N}(0,1)$ (Proposition 4.2). The proof relies on extending the idea of the previous paragraph that any non-negative function $f$ which vanishes (i.e. gives value zero) on the $k'$ points in question satisfies $\frac{\mathbf{E}_{x\sim A}[f^2(x)]}{\mathbf{E}_{x\sim A}[f(x)]^2} \leq 1/w_0^2$. We specifically use $f(x) = x^t p(x)$, where $p(x) = (x - \mu_1)^2 \ldots (x - \mu_{k'})^2$ and $\mu_1, \ldots, \mu_{k'}$ are the points in the support of $A$ with unrestricted weights. The goal then is to show that the ratio $r := \frac{\mathbf{E}_{x\sim A}[f^2(x)]}{\mathbf{E}_{x\sim A}[f(x)]^2}$ is at least $2^{\Omega(t-k')}$ — combining this with our earlier lower bound $r \leq 1/w_0^2$ implies $t = O(\log(1/w_0)+k')$. To bound $r$, we assume $A$ matches $\Theta(t + k')$ moments, allowing us to replace $\mathbf{E}_{x\sim A}[\cdot]$ with $\mathbf{E}_{x\sim\mathcal{N}(0,1)}[\cdot]$ in the definition of $r$. Since $p(x)$ has degree $2k'$, we show $p(x) \geq 2^{-O(k')}\mathbf{E}_{x\sim\mathcal{N}(0,1)}[p^2(x)]^{1/2}$

near $x = \sqrt{2t}$ (Corollary 4.5). The contribution to $\mathbf{E}_{x \sim \mathcal{N}(0,1)}[f^2(x)]$ from $x \in [0.9\sqrt{2t}, 1.1\sqrt{2t}]$ will then be at least $(3.6t)^t \mathbf{E}_{x \sim \mathcal{N}(0,1)}[p^2(x)]2^{-O(k')}$ (see Equations (7) and (8) for the full calculations). Meanwhile, by Hölder's inequality, $\mathbf{E}_{x \sim \mathcal{N}(0,1)}[f(x)] \leq \mathbf{E}_{x \sim \mathcal{N}(0,1)}[x^{3t/2}]^{2/3} \mathbf{E}_{x \sim \mathcal{N}(0,1)}[p^3(x)]^{1/3}$, which by hypercontractivity is at most $(\frac{3t}{2e})^{t/2} \mathbf{E}_{x \sim \mathcal{N}(0,1)}[p(x)^2]^{1/2}2^{O(k')}$. Combining these bounds establishes $r \geq 2^{\Omega(t-k')}$ and proves Proposition 4.2.

We can also argue that the previous paragraph's result can always be used with $w_0 \geq 2^{-O(k')}/k$ (Proposition 4.1). By considering the same polynomial $p$ which vanishes at the points with unconstrained weights, we can combine the hypercontractivity of $p$ with the Cauchy-Schwarz inequality to derive a lower bound on the total weight of the equal-weight points: $\sum_{i \geq k'+1} w_i \geq 2^{-O(k')}$. This immediately implies $w_0 \geq 2^{-O(k')}/k$.

We have shown that any discrete distribution with $k'$ arbitrary weights and $k - k'$ equal weights cannot match more than $O(\log(k) + k')$ moments with $\mathcal{N}(0,1)$. This result extends to approximate moment matching within error $2^{-O(t)}$, and holds even after convolving the distribution with a Gaussian (cf. Lemma D.3). For the parallel pancakes testing problem, this implies that for some $i \leq O(\log(k) + k')$ the $i$-th order tensor of the GMM in the alternative hypothesis differs significantly from that of $\mathcal{N}(0, I)$. This gap can be detected by estimating the tensor via averaging samples (an operation that has complexity $d^{\Theta(m)}$), leading to the testing algorithm of Theorem 1.4.

## 2. Preliminaries

We present only the essential preliminaries here; see Appendix B for a full version.

**Notation** We use $\mathbb{Z}_+$ for positive integers, $\mathbb{R}_0^+$ for non-negative reals, and $[n] \stackrel{\text{def}}{=} \{1, \ldots, n\}$. We use $x \otimes y$ for the tensor product of two vectors. For a random variable $x$ following distribution $\mathcal{D}$, we write $x \sim \mathcal{D}$ and $\mathbf{E}[x]$ for its expectation. The Gaussian distribution with mean $\mu$ and covariance $\Sigma$ is $\mathcal{N}(\mu, \Sigma)$, and $\mathbf{Pr}(\mathcal{E})$ denotes the probability of event $\mathcal{E}$. The indicator function of $\mathcal{E}$ is $\mathbf{1}(\mathcal{E})$. The $L_p$ norm of an $\mathbb{R}$-valued random variable $x$ is $\|x\|_p = \mathbf{E}[|x|^p]^{1/p}$, and for a function $f : \mathbb{R}^d \to \mathbb{R}$, it is $\|f\|_p = \mathbf{E}_{x \sim \mathcal{N}(0,I)}[|f(x)|^p]^{1/p}$. We use $a \lesssim b$ to indicate $a \leq Cb$ for an absolute constant $C > 0$ independent of $a$ and $b$.

**Hermite Analysis** In this paper, we use the *normalized probabilist's* Hermite polynomials, which form an orthonormal basis of $L^2 := \{f : \mathbf{E}_{x \sim \mathcal{N}(0,1)}[f^2(x)] < \infty\}$ with respect to the Gaussian measure, i.e., $\int_{\mathbb{R}} h_k(x)h_m(x)e^{-x^2/2} dx = \sqrt{2\pi}\mathbf{1}(k = m)$. Every function $f \in L^2$ can be uniquely expressed as $f(x) = \sum_{i=0}^{\infty} a_i h_i(x)$.

**Probability Facts** The first fact below follows from direct calculations, the second from the Carbery-Wright inequality, and the last from Hölder's inequality combined with Fact 2.3.

**Fact 2.1** (Gaussian Moments). $\mathbf{E}_{x \sim \mathcal{N}(0,1)}[x^t] \lesssim (t/e)^{t/2} \ \forall t \geq 0$.

**Fact 2.2.** *For every polynomial of degree $r$ and every $\epsilon > 0$,* $\mathbf{Pr}_{x \sim \mathcal{N}(0,1)}(|p(x)| \leq \epsilon\|p\|_1) \leq O(r\epsilon^{1/r})$.

**Fact 2.3** (Gaussian Hypercontractivity). *If $p$ is a degree $r$ polynomial and $k > 2$, then $\|p\|_k \leq (k-1)^{r/2}\|p\|_2$.*

**Fact 2.4.** *For any polynomial $p$ of degree $r$, $\frac{\|p\|_1}{\|p\|_2} \geq 3^{-r}$.*

**Arithmetic Mean-Geometric Mean Inequality** We record the following continuous analog of the *Arithmetic Mean-Geometric Mean* (AM-GM) inequality. We refer to Appendix B for a more detailed discussion.

**Fact 2.5** (Continuous AM-GM Inequality). *Let $f : \mathbb{R} \to \mathbb{R}_0^+$ be a function, and let $I \subseteq \mathbb{R}$ be a finite interval. If $f(x)$ and $\ln f(x)$ are integrable on $I$, then the following holds:* $\frac{1}{|I|} \int_I f(x)\mathrm{d}x \geq \exp\left(\frac{1}{|I|} \int_I \ln f(x)\mathrm{d}x\right)$.

**Non-Gaussian Component Analysis** The parallel pancakes Problem 1.1 is a special case of the following problem.

**Problem 2.6** (Non-Gaussian Component Analysis (NGCA)). Let $B$ be a distribution on $\mathbb{R}$. For a unit vector $v$, we denote by $P_{B,v}$ the distribution with the density $P_{B,v}(x) := B(v^\top x)\phi_{\perp v}(x)$, where $\phi_{\perp v}(x) = \exp\left(-\|x - (v^\top x)v\|_2^2/2\right)/(2\pi)^{(d-1)/2}$, i.e., the distribution that coincides with $B$ on the direction $v$ and is standard Gaussian in every orthogonal direction. We define the following hypothesis testing problem:

- $H_0$: The data distribution is $\mathcal{N}(0, I_d)$.
- $H_1$: The data distribution is $P_{B,v}$, for some vector $v \in \mathcal{S}^{d-1}$ in the unit sphere.

It is known that solving Problem 2.6 when $B$ matches the first $m$ moments with $\mathcal{N}(0,1)$ requires at least $d^{\Omega(m)}$ complexity in the statistical query model (Proposition B.8).

## 3. The Uniform Weights Case

In this section we prove the following proposition, which is sufficient for showing our first result, Theorem 1.3.

**Proposition 3.1.** *For each $k$ that is larger than a sufficiently large absolute constant, there exists a set $S$ of $k$ points in $\mathbb{R}$ such that the uniform distribution over $S$ matches the first $\Omega(\log k)$ moments with $\mathcal{N}(0,1)$.*

Given the above, Theorem 1.3 follows directly from standard SQ theory. The details are provided in Appendices B.5

and C, but the steps are summarized as follows: Let $A$ be the uniform distribution on the set $S$ from Proposition 3.1. We can define the distribution $B$ to be what one obtains by first drawing a sample from $A$, rescaling it by $1/\sqrt{\delta}$ and adding Gaussian noise $\mathcal{N}(0, 1 - \delta)$. This operation preserves moment matching and makes $B$ a GMM. The NGCA Problem 2.6 with that $B$ then becomes equivalent to the parallel pancakes Problem 1.1. Since $B$ matches $m = \Omega(\log k)$ moments with $\mathcal{N}(0,1)$, its standard SQ hardness state that its complexity is $d^{\Omega(m)} = d^{\Omega(\log k)}$ (Proposition B.8). We refer to Appendix C for the details of this paragraph.

In the remainder, we focus on proving Proposition 3.1 by leveraging a result on designs theory from Kane (2015). The original result in Kane (2015) is highly general and applies to a wide range of topological, path-connected design problems. However, as we will only use the theorem for intervals, we present here a specialized version tailored to this case.

**Fact 3.2** (see Theorem 4 in Kane (2015)). *Let $t \in Z_+$ be an integer, $I \subset \mathbb{R}$ be an interval and $Q$ be a distribution on $I$. Let $W_t$ be the vector space of all polynomials of degree at most $t$, and $V_t$ be the vector space of polynomials $p$ of degree at most $t$ with $\mathbf{E}_{x \sim Q}[p(x)] = 0$. Define $K_t = \sup_{p \in V \setminus \{0\}} \frac{\sup_{x \in I} p(x)}{|\inf_{x \in I} p(x)|}$. Then for every integer $n > (t-1)(K_t + 1)$ there exists a set $S \subset I$ of $n$ points such that $\frac{1}{|S|} \sum_{x \in S} p(x) = \mathbf{E}_{x \sim Q}[p(x)]$ for all $p \in W_t$.*

Our goal is to show that $K_t = 2^{O(t)}$ for $Q = \mathcal{N}(0,1)$, which would directly imply Theorem 1.3. However, as noted in Section 1.2, $K_t$ may be infinite when $I = \mathbb{R}$. To address this, we use a distribution $Q$ supported on a bounded interval of $\mathbb{R}$ that matches the first $t$ moments of $\mathcal{N}(0,1)$. Applying Fact 3.2 with this $Q$ also suffices for establish Theorem 1.3.

**Lemma 3.3** (Gaussian Quadrature (Lemma 4.3 in Diakonikolas et al. (2017))). *There is a discrete distribution $Q$ on the real line, supported on $t$ points, that agrees with $\mathcal{N}(0,1)$ on the first $2t - 1$ moments. All points $x$ in the support of $Q$ have $|x| = O(\sqrt{t})$.*

We start with an anti-concentration property of Gaussian polynomials that will be useful for bounding the numerator in the definition of $K_t$.

**Lemma 3.4.** *Let $C$ be a sufficiently large constant. For every polynomial $p : \mathbb{R} \to \mathbb{R}$ of degree at most $t$ with $\mathbf{E}_{x \sim \mathcal{N}(0,1)}[p(x)] = 0$ and for every $\epsilon > 0$ it holds*

$$\Pr_{x \sim \mathcal{N}(0,1)}(p(x) > \epsilon \|p\|_1) \geq \left( \frac{1}{2} \frac{\|p\|_1}{\|p\|_2} (1 - Ct\epsilon^{1+1/t}) \right)^2 .$$

*Proof.* Denote by $\phi(x)$ the pdf of $\mathcal{N}(0,1)$. We have the following (each step is explained below):

$$\|p\|_1 = \int_{p(x)>0} p(x)\phi(x)\mathrm{d}x - \int_{p(x)\leq 0} p(x)\phi(x)\mathrm{d}x = 2\int_{p(x)>0} p(x)\phi(x)\mathrm{d}x$$

$$= 2 \left( \int_{p(x)\geq\epsilon\|p\|_1} p(x)\phi(x)\,\mathrm{d}x + \int_{0\leq p(x)<\epsilon\|p\|_1} p(x)\phi(x)\mathrm{d}x \right)$$

$$\leq 2\|p\|_2 \Pr_{x \sim \mathcal{N}(0,1)}(p(x) \geq \epsilon\|p\|_1)^{1/2}$$

$$+ 2\epsilon\|p\|_1 \Pr_{x \sim \mathcal{N}(0,1)}(|p(x)| \leq \epsilon\|p\|_1)$$

$$\leq 2\|p\|_2 \Pr_{x \sim \mathcal{N}(0,1)}(p(x) \geq \epsilon\|p\|_1)^{1/2} + 2\epsilon\|p\|_1 Ct\epsilon^{1/t},$$

where the first line used that $\mathbf{E}_{x \sim \mathcal{N}(0,1)}[p(x)] = 0$, the penultimate inequality used the Cauchy–Schwarz inequality for the first term and the bound $p(x) \leq \epsilon\|p\|_1$ for the second term, and the last line used the Carbery-Wright inequality (Fact 2.2). Rearranging, we obtain $\sqrt{\mathbf{Pr}_{x \sim \mathcal{N}(0,1)}(p(x) > \epsilon\|p\|_1)} \geq \frac{1}{2}\frac{\|p\|_1}{\|p\|_2}(1 - 2Ct\epsilon^{1+1/t})$. We rename the constant $2C$ to $C$. $\square$

We now bound $K_t$ from Fact 3.2 with $I = [-C\sqrt{t}, C\sqrt{t}]$.

**Lemma 3.5.** *Let $t > C$ be an integer, where $C$ is a sufficiently large constant, and define $I = [-C\sqrt{t}, +C\sqrt{t}]$. For every polynomial $p$ of degree at most $t$ with $\mathbf{E}_{x \sim \mathcal{N}(0,1)}[p(x)] = 0$ it holds $\frac{\sup_{x \in I} p(x)}{|\inf_{x \in I} p(x)|} \leq 2^{O(t)}$.*

*Proof.* It suffices to upper bound the numerator by $2^{O(t)}\|p\|_1$ and lower bound the denominator by $2^{-O(t)}\|p\|_1$.

**Upper bound on numerator** We require the following:

**Fact 3.6** (Krasikov (2004)). *For the $k$-th normalized probabilist's Hermite polynomial $h_k$, we have $\sup_{x \in \mathbb{R}} h_k^2(x)e^{-x^2/2} = O(k^{-1/6})$.*

Consider a polynomial $p$ which has degree at most $t$ and satisfies $\mathbf{E}_{x \sim \mathcal{N}(0,1)}[p(x)] = 0$. We first expand the polynomial in the Hermite basis: $p(x) = \sum_{k=1}^{t} a_k h_k(x)$, where the summation starts from $k = 1$ because $a_0 = \mathbf{E}_{x \sim \mathcal{N}(0,1)}[p(x)] = 0$. For any $x \in I$ we have (the first step uses Cauchy–Schwarz inequality):

$$|p(x)| = \left| \sum_{k=1}^{t} a_k h_k(x) \right| \leq \sqrt{\sum_{k=1}^{t} a_k^2} \sqrt{\sum_{k=1}^{t} h_k^2(x)}$$

$$\lesssim \|p\|_2 \sqrt{\sum_{k=1}^{t} e^{x^2/2} k^{-1/6}} \qquad \text{(by Fact 3.6)}$$

$$\leq \|p\|_2 2^{O(t)} \sqrt{\sum_{k=1}^{t} k^{-1/6}} \quad \text{(using } |x| = O(\sqrt{k}))$$

$$\leq \|p\|_2 2^{O(t)} t^{O(1)} \quad \text{(using } \sum_{k=1}^{t} k^{-1/6} = t^{O(1)})$$

$$\leq \|p\|_2 2^{O(t)} \leq \|p\|_1 2^{O(t)} . \qquad \text{(using Fact 2.4)}$$

**Lower bound on the denominator** From Lemma 3.4 with

$-p$ in place of $p$ and $\epsilon = 2^{-t}$, and Fact 2.4 we get that

$$\Pr_{x \sim \mathcal{N}(0,1)} (p(x) < -2^{-t} \|p\|_1) \geq \left( \frac{1}{2} \frac{\|p\|_1}{\|p\|_2} (1 - Ct2^{-t-1}) \right)^2$$

$$\geq \left( \frac{1}{2} 3^{-t} (1 - Ct2^{-t-1}) \right)^2 > 2^{-4t} .$$

where we used that $t$ is big enough so that $C \frac{t}{2^t} < 0.5$. Then,

$$\Pr_{x \sim \mathcal{N}(0,1)} (p(x) < -2^{-t} \|p\|_1, x \in I)$$

$$\geq \Pr_{x \sim \mathcal{N}(0,1)} (p(x) < -2^{-t} \|p\|_1) - \Pr_{x \sim \mathcal{N}(0,1)} (x \notin I)$$

$$\geq 2^{-4t} - 2^{-100t} > 0 ,$$

where in the last line we used that $I = [-C\sqrt{t}, +C\sqrt{t}]$ for a large constant $C$. We have thus shown that $\inf_{x \in I} p(x) \leq -2^{-t} \|p\|_1$ and therefore $|\inf_{x \in I} p(x)| > 2^{-t} \|p\|_1$. $\qquad \square$

## 4. The Mostly Equal Weights Case

This section focuses on Theorem 1.4 and is organized as follows. The key structural result is the following impossibility of moment matching: if $A$ is a distribution on $k$ points, with $k'$ points having unconstrained weights and the remaining $k - k'$ equal, then $A$ cannot match more than $O(\log k + k')$ moments with the standard Gaussian.

**Proposition 4.1.** *Let $k' < k$ be positive integers, and let $A$ be a discrete distribution on $k$ points in $\mathbb{R}$. Suppose $k - k'$ of the points have equal probability masses, while the remaining $k'$ points have unrestricted probability masses. Denote by $m$ the highest degree for which every degree-$m' \leq m$ polynomial $g$ satisfies $\left| \mathbf{E}_{x \sim A}[g(x)] - \mathbf{E}_{x \sim \mathcal{N}(0,1)}[g(x)] \right| \leq 2^{-C \cdot m} \|g\|_2$, then $m$ must satisfy $m \leq O(\log k) + O(k')$.*

Section 4.1 explains how Proposition 4.1 leads to a testing algorithm (the full proof appears in Appendix D.1). Section 4.2 provides the proof of Proposition 4.1.

### 4.1. Proof Sketch of Theorem 1.4

Consider the parallel pancakes problem from Theorem 1.4, which is equivalent to the NGCA problem (Problem 2.6) with the 1-d GMM $B = \sum_{i \in [k]} w_i \mathcal{N}(\mu_i, 1-\delta)$. If $B$ approximately matches $m$ moments of $\mathcal{N}(0,1)$, we aim to show that $m \leq O(\log k + k')$, enabling a testing algorithm that detects significant deviations in moment tensors. Specifically, suppose every polynomial $p$ of degree $m' \leq m$ with $\|p\|_2 = 1$ satisfies $|\mathbf{E}_{x \sim B}[p(x)] - \mathbf{E}_{x \sim \mathcal{N}(0,1)}[p(x)]| \leq (\delta/2)^{Cm}$ for some large constant $C \gg 1$. Now, consider the discrete distribution $A$, which assigns weight $w_i$ to each center $\mu_i/\sqrt{\delta}$. By Lemma D.3, $A$ also approximately matches the moments of $\mathcal{N}(0,1)$, but with an error of $2^{-O(m)}$ instead of $(\delta/2)^{O(m)}$. Then Proposition 4.1 yields $m \leq O(\log k + k')$, as desired.

We just showed that there is a polynomial $p$ of degree at most $m = O(\log(k) + k')$, where the expectations under $B$ and $\mathcal{N}(0,1)$ differ significantly: $\lambda := \left| \mathbf{E}_{x \sim B}[p(x)] - \mathbf{E}_{x \sim \mathcal{N}(0,1)}[p(x)] \right| > (\delta/2)^{Cm}$. An averaging argument further implies that a gap holds even for some monomial $x^i$. Lifting this to the $d$-dimensional parallel pancakes, we have the moment tensor gap $\mathbf{E}_{x \sim P_{B,v}}[x^{\otimes i}] - \mathbf{E}_{x \sim \mathcal{N}(0,I)}[x^{\otimes i}] = \pm \lambda v^{\otimes i}$.

The Frobenius norm of the gap is $\lambda > (\delta/2)^{Cm}$, implying that between the (expected) moment tensors, at least one entry differs by at least $\epsilon := \lambda/d^m = (d/\delta)^{(C-1)m}$. We will test by searching for such an entry in the empirical tensor.

---

**Algorithm 1** Testing Algorithm (simplified)

---

1: **Input**: $k, n$. **Output**: $\hat{H} \in \{H_0, H_1\}$.

2: **For** $i = 1, 2, 3, \ldots, C \cdot (\log(k) + k')$ **do**
3:      Draw $x_1, \ldots, x_n \sim D$.
4:      Define $M \leftarrow \frac{1}{n} \sum_{i=1}^{n} x^{\otimes i}$.
5:      Define $M' := \mathbf{E}_{x \sim \mathcal{N}(0,I)}[x^{\otimes i}]$.
6:      **If** $\exists a = (i_1, \ldots, j_i)$ such that $|M_a - M'_a| > d^{-Cm} \lambda_m$
7:          **then** Output $H_1$ and terminate.
8: **Return** $H_0$.

---

The tester above is a simplified version. However, it is not fully correct, as we must ensure the concentration of the empirical tensor to bound the sample complexity. The Gaussian $\mathcal{N}(0,I)$'s empirical tensor is well-concentrated. While the parallel pancake's tensor might not concentrate well, this happens only when there is a Gaussian component much farther than $\sqrt{d}$ from the origin — otherwise every sample from the parallel pancake is within $O(\sqrt{d})$ in norm in high probability, and the empirical tensor is entrywise well-concentrated (e.g. by Hoeffding). This is also a testable condition: with $\gg \log(k)/w_{\min}$ samples, we will be able to check if every component is centered at most $O(\sqrt{d})$ from the origin. The full version of the algorithm with this additional preliminary check, along with its correctness proof, are provided in Appendix D.1.

### 4.2. Proof of Proposition 4.1

We now show Proposition 4.1. We will actually show a slightly different version below, where $k'$ of the points have arbitrary weights and the rest have weight at least $w_0$.

**Proposition 4.2.** *Let $C$ be a sufficiently large constant, and let $k' < k$ be positive integers. Let $A$ be a discrete distribution on $k$ points in $\mathbb{R}$ with probability masses $w_1, \ldots, w_k$, where $w_i \geq w_0$ for $i = k'+1, \ldots, k$ (i.e., the last $k - k'$ weights are lower bounded by $w_0$, while the first $k'$ weights are unrestricted). Let $m$ be the largest degree such that every polynomial $g$ of degree $m' \leq m$ satisfies*

$$\left| \mathbf{E}_{x \sim A}[g(x)] - \mathbf{E}_{x \sim \mathcal{N}(0,1)}[g(x)] \right| \leq w_0 2^{-C \cdot m} \|g\|_2 . \quad (1)$$

*Then $m \le O(\log(1/w_0)) + O(k')$.*

Proposition 4.1 can be derived from this via the following observation (shown in Appendix D.2): in the setting of Proposition 4.2, let $p(x) = \prod_{i=1}^{k'} (x - \mu_i)$, where $\mu_1, \ldots, \mu_{k'}$ are the points in the support of $A$ with the unconstrained weights. Then, the weights of the $k - k'$ points with uniform weights is always $\sum_{i=k'+1}^{k} w_i \ge \frac{\mathbf{E}_{x \sim A}[p(x)]^2}{\mathbf{E}_{x \sim A}[p^2(x)]} \gtrsim \frac{\|p\|_1^2}{\|p\|_2^2} \ge 3^{-2k'}$, where the first step uses Cauchy-Schwarz inequality, the second uses the (approximate) moment matching, and the third is a consequence of Gaussian hypercontractivity (Fact 2.4). This means that every such weight is $w_i \ge 3^{-2k'}/k$, which when plugged into Proposition 4.2 gives Proposition 4.1.

We now focus on showing Proposition 4.2. We will follow a top-down presentation, starting with the proof strategy and concluding with a derivation of the necessary bounds.

We will use a reparameterization $m = 2t + 4k'$ with $t$ even. The goal is to show that if $A$ is assumed to approximately match the first $m = 2t + 4k'$ moments with $\mathcal{N}(0,1)$ (in the sense of Equation (1)), then $t$ must be at most $O(\log(1/w_0) + k')$. Let $\mu_1, \ldots, \mu_k$ be the points on which $A$ is supported, where the first $k'$ points are the ones with the unrestricted weights, and consider the polynomial $f(x) = x^t p(x)$, where $p(x) = (x - \mu_1)^2 (x - \mu_2)^2 \cdots (x - \mu_{k'})^2$. The proof strategy is the following: if the expectation of $f$ under $A$ approximately matches that of $\mathcal{N}(0,1)$, then the value of $f$ on every point $\mu_i$ cannot be too large, which will cause the expectations of $f^2$ to deviate.

Because of Equation (1) with $g(x) = f^2(x)$, we have:

$$\sum_{i=k'+1}^{k} w_i \, \mu_i^t \, p(\mu_i) = \mathbf{E}_{x \sim A}\left[x^t p(x)\right] \qquad (2)$$

$$\le \mathbf{E}_{x \sim \mathcal{N}(0,1)}\left[x^t p(x)\right] + w_0 \frac{\|x^t p(x)\|_2}{2^{C(2t+4k')}} . \qquad (3)$$

This, together with the lower bound $w_i \ge w_0$ for the points $i = k'+1, \ldots, k$ and the fact that $t$ is even, implies that for all $i = k'+1, \ldots, k$ it holds

$$\mu_i^t p(\mu_i) \le \frac{1}{w_0} \mathbf{E}_{x \sim \mathcal{N}(0,1)}\left[x^t p(x)\right] + \frac{\|x^t p(x)\|_2}{2^{C(2t+4k')}} . \qquad (4)$$

We now examine the expectations of the square of $f(x)$. Because of Equation (1) with $g(x) = f^2(x)$, we have

$$\mathbf{E}_{x \sim \mathcal{N}(0,1)}\left[x^{2t} p^2(x)\right] \le \mathbf{E}_{x \sim A}\left[x^{2t} p^2(x)\right] + \frac{\|x^{2t} p^2(x)\|_2}{2^{C(2t+4k')}}$$

$$= \sum_{i=k'+1}^{k} w_i \left(\mu_i^t p(\mu_i)\right)^2 + \frac{\|x^{2t} p^2(x)\|_2}{2^{C(2t+4k')}} .$$

Next, we can combine this with Equation (4), divide both sides by $\mathbf{E}_{x \sim \mathcal{N}(0,1)}\left[x^t p(x)\right]^2$ (and use $\sum_{i=k'+1}^{k} w_i \le 1$) to obtain the following, where $\lambda := 2^{-C(2t+4k')}$:

$$\frac{\mathbf{E}_{x \sim \mathcal{N}(0,1)}\left[x^{2t} p^2(x)\right]}{\mathbf{E}_{x \sim \mathcal{N}(0,1)}\left[x^t p(x)\right]^2} \le \left(\frac{1}{w_0}\right)^2$$

$$+ \lambda^2 \frac{\mathbf{E}_{x \sim \mathcal{N}(0,1)}\left[x^{2t} p^2(x)\right]}{\mathbf{E}_{x \sim \mathcal{N}(0,1)}\left[x^t p(x)\right]^2} + \lambda \frac{\mathbf{E}_{x \sim \mathcal{N}(0,1)}\left[x^{4t} p^4(x)\right]^{\frac{1}{2}}}{\mathbf{E}_{x \sim \mathcal{N}(0,1)}\left[x^t p(x)\right]^2} .$$

Let us simplify this inequality. Let $r$ be the ratio on the LHS. The second term on the RHS is $\lambda^2 \cdot r$. The third term is at most $3^{t+2k'} \lambda r$, by applying Gaussian hypercontractivity (Fact 2.3) to the polynomial $x^t p(x)$. Thus, the inequality becomes $r(1 - \lambda^2 - \lambda 3^{t+2k'}) \lesssim 1/w_0^2$. Since $\lambda = 2^{-C(2t+4k')}$ for large constant $C$, the expression inside the parentheses is greater than 0.5. Therefore, the inequality implies that $r \lesssim 1/w_0^2$.

The next step is to establish a lower bound for $r$, specifically to show that $r \ge 2^{\Omega(t)}/2^{O(k')}$. If this can be done, combining the two bounds $2^{\Omega(t)}/2^{O(k')} \le 1/w_0^2$ and taking logarithms yields $t = O(\log(1/w_0)) + O(k')$, completing the proof of Proposition 4.2.

### 4.2.1. LOWER BOUNDING THE RATIO $r$

We want to establish the following, which was the missing piece in the proof of Proposition 4.2 above.

**Lemma 4.3.** *Let $p : \mathbb{R} \to \mathbb{R}_0^+$ be a polynomial of the form $p(x) = (x - \mu_1)^2 (x - \mu_2)^2 \cdots (x - \mu_{k'})^2$. Then*

$$\frac{\mathbf{E}_{x \sim \mathcal{N}(0,1)}\left[x^{2t} p^2(x)\right]}{\mathbf{E}_{x \sim \mathcal{N}(0,1)}\left[x^t p(x)\right]^2} \gtrsim \frac{2^{\Omega(t)}}{2^{O(k')}} .$$

The most difficult part involves lower bounding the numerator. To this end, we will show the following bound:

**Lemma 4.4.** *Let $p : \mathbb{R} \to \mathbb{R}$ be a polynomial of the form $p(x) = (x - \mu_1)(x - \mu_2) \cdots (x - \mu_{k'})$ where $\mu_1, \ldots, \mu_{k'} \in \mathbb{R}$, and define $I := [0.9\sqrt{2t}, 1.1\sqrt{2t}]$. For every $t > 0$ and for every $\mu_1, \ldots, \mu_{k'} \in \mathbb{R}$, the following holds:*

$$\exp\left(\frac{1}{|I|} \int_{x \in I} \ln|p(x)|\mathrm{d}x\right) \ge \max_{y \in \mathbb{R} : |y| \le \sqrt{t}} \frac{|p(y)|}{2^{O(k')}} . \qquad (5)$$

We will actually apply Lemma 4.4 after taking expectations of both sides. This version is presented below, and its proof follows by taking expectations and performing some manipulations (see Appendix D.3 for a detailed proof).

**Corollary 4.5.** *Let $p : \mathbb{R} \to \mathbb{R}$ be a polynomial of the form $p(x) = (x - \mu_1)(x - \mu_2) \cdots (x - \mu_{k'})$ where $\mu_1, \ldots, \mu_{k'} \in \mathbb{R}$ are arbitrary parameters. Define $I = [0.9\sqrt{2t}, 1.1\sqrt{2t}]$. For all $t \ge 1$ we have $\exp\left(\frac{1}{|I|} \int_{x \in I} \ln|p(x)|\mathrm{d}x\right) \ge \frac{\|p\|_2}{2^{O(k')}}$.*

To see why the above bound is needed to prove Lemma 4.3, we will first prove Lemma 4.3 assuming Corollary 4.5. Then, we will prove Lemma 4.4.

*Proof of Lemma 4.3.* First, for the denominator, we have the following:

$$\mathop{\mathbf{E}}_{x \sim \mathcal{N}(0,1)}[x^t p(x)] \leq \mathop{\mathbf{E}}_{x \sim \mathcal{N}(0,1)}[p^3(x)]^{1/3} \mathop{\mathbf{E}}_{x \sim \mathcal{N}(0,1)}[x^{3t/2}]^{2/3}$$

$$\lesssim \|p\|_3 \left(\frac{3t}{2e}\right)^{t/2} \lesssim 2^{k'} \|p\|_2 \left(\frac{3t}{2e}\right)^{t/2}, \quad (6)$$

where the first step uses Hölder's inequality, the second step uses the Gaussian moments bound (Fact 2.1), and the final step uses Gaussian hypercontractivity (Fact 2.3).

We now lower bound the numerator. Define $I := [0.9\sqrt{2t}, 1.1\sqrt{2t}]$. We have the following (see below for explanations of each step):

$$\mathop{\mathbf{E}}_{x \sim \mathcal{N}(0,1)}\left[x^{2t} p^2(x)\right] \gtrsim \int_{-\infty}^{+\infty} x^{2t} e^{-x^2/2} p^2(x)\, \mathrm{d}x$$

$$\geq \int_{x \in I} x^{2t} e^{-x^2/2} p^2(x)\, \mathrm{d}x$$

$$\geq (1.62t)^t e^{-0.81t} \int_{x \in I} p^2(x)\, \mathrm{d}x$$

$$= (1.62t)^t e^{-0.81t} |I| \left(\frac{1}{|I|} \int_{x \in I} p^2(x)\, \mathrm{d}x\right)$$

$$\gtrsim (1.62t)^t e^{-0.81t} \left(\frac{1}{|I|} \int_{x \in I} p^2(x)\, \mathrm{d}x\right), \quad (7)$$

where the third inequality uses that $\min_{x \in I} x^{2t} e^{-x^2/2} \geq (1.62t)^t e^{-0.81t}$, and the final inequality uses that $|I| = 0.2\sqrt{2t} = \Omega(1)$. We now focus on the root mean square term $\frac{1}{|I|} \int_{x \in I} p^2(x)\, \mathrm{d}x$, which we will bound using the AM-GM inequality (Fact 2.5) and the geometric mean bound from Lemma 4.4. The first step below applies Fact 2.5 with $f(x) := p^2(x)$, and the next step uses Corollary 4.5.

$$\frac{1}{|I|} \int_{x \in I} p^2(x)\, \mathrm{d}x \geq \exp\left(\frac{1}{|I|} \int_{x \in I} \ln |p(x)|\, \mathrm{d}x\right)^2 \geq \frac{\|p\|_2^2}{2^{O(k')}}.$$

Combining with Equation (7), we obtain the following bound for the numerator:

$$\mathop{\mathbf{E}}_{x \sim \mathcal{N}(0,1)}\left[x^{2t} p^2(x)\right] \gtrsim (1.62t)^t e^{-0.81t} \frac{\|p\|_2^2}{2^{O(k')}}. \quad (8)$$

Combining Equation (6) and Equation (8), we conclude

$$\frac{\mathbf{E}_{x \sim \mathcal{N}(0,1)}\left[x^{2t} p^2(x)\right]}{\mathbf{E}_{x \sim \mathcal{N}(0,1)}\left[x^t p(x)\right]^2} \gtrsim \frac{(1.62)^t}{\left(\frac{1.5}{e}\right)^t e^{0.81t} 2^{O(k')}} \geq \frac{(1.3)^t}{2^{O(k')}}. \ \square$$

We conclude this section by proving Lemma 4.4.

*Proof of Lemma 4.4.* Fix an arbitrary $y \in \mathbb{R}$ with $|y| \leq \sqrt{t}$. First, note that by the property of logarithms and sums, we can write the left hand side as

$$\exp\left(\sum_{i=1}^{k'} \frac{1}{|I|} \int_{x \in I} \ln |x - \mu_i|\, \mathrm{d}x\right).$$

In order to show Equation (5), it suffices to work with each term and show the following for each $i \in [k']$:

$$\frac{1}{|I|} \int_{x \in I} \ln |x - \mu_i| \geq \ln |y - \mu_i| - O(1).$$

Equivalently, it suffices to show that Equation (5) holds for every linear polynomial of the form $p(x) = x - a$. Therefore, the goal for the rest of this proof is to show that

$$\exp\left(\frac{1}{|I|} \int_{x \in I} \ln |x - a|\, \mathrm{d}x\right) \geq |y - a|/O(1), \quad (9)$$

holds for every $a \in \mathbb{R}$ and $y \in \mathbb{R}$ with $|y| \leq \sqrt{t}$. We will examine two cases.

**Case 1** The first case is when the root $a$ of the polynomial is outside the interval $I$. In this case, we can show that $|x - a|/|y - a| = \Theta(1)$, which implies $\ln |x - a| \geq \ln |y - a| - O(1)$, and the desired conclusion (Equation (9)) follows by integrating both sides and applying the $\exp(\cdot)$ function.

To show the earlier claim that $|x - a|/|y - a| = \Theta(1)$, we can consider the following sub-cases:

1. Case $a \geq 1.1\sqrt{2t}$ (i.e., $a$ is to the right of $I$): Suppose $a = 1.1\sqrt{2t} + u$ for some non-negative $u$. Then, $a - x = (1.1\sqrt{2t} - x) + u = \Theta(\sqrt{t}) + u$ and $a - y = (1.1\sqrt{2t} - y) + u = \Theta(\sqrt{t}) + u$. Therefore, for any $u \geq 0$, the ratio $|x - a|/|y - a| = (\Theta(\sqrt{t}) + u)/(\Theta(\sqrt{t}) + u) = \Theta(1)$.

2. The cases $a < -\sqrt{t}$ and $a \in [\sqrt{t}, 0.9\sqrt{2t}]$ can be shown in a similar manner.

**Case 2** Suppose that the root $a$ of the polynomial $p$ lies within the interval $I$. In that case, we can show via derivative analysis that $f(a) := \frac{1}{|I|} \int_{x \in I} \ln |x - a|\, \mathrm{d}x$ for $a \in I$ is minimized at the midpoint of $I$, i.e., at $a = \sqrt{2t}$, and confirm that $f(\sqrt{2t}) \geq \sqrt{t}/20 = \Omega(|y - a|)$. These calculations are provided in Appendix D.3. $\square$

## Conclusions and Future Work

Our work makes progress in understanding the complexity of learning parallel pancake GMMs, in terms of both lower and upper bounds. We establish the tightness of existing algorithms for uniform weights and provide an improved testing algorithm for uneven weights. A number of interesting open problems remain:

- Can we extend our testing algorithm to learning the unknown direction of the parallel pancakes? More broadly, can we characterize the complexity of learning GMMs with common covariance and not necessarily collinear means as a function of the weights distribution?

- Can we obtain an algorithm with quasi-polynomial (i.e., $d^{O(\log(1/w_{\min}))}$) complexity for GMMs whose components have unknown (and potentially different) covariances?

## Impact Statement

This work is theoretical in nature and focuses on advancing fundamental knowledge. As such, it does not directly raise any societal or ethical concerns that warrant special consideration.

## Acknowledgments

Ilias Diakonikolas was supported by NSF Medium Award CCF-2107079 and an H.I. Romnes Faculty Fellowship. Daniel M. Kane was supported by NSF Medium Award CCF-2107547 and NSF Award CCF-1553288 (CAREER). The work of Jasper C.H. Lee was done in part while he was at UW Madison, supported by NSF Medium Award CCF-2107079.

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

# Supplementary Material

**Organization** Appendix A discusses additional related work on Gaussian mixture models, Appendix B provides the full version of the preliminaries needed for the technical proofs, Appendix C provides the missing details from the proof of our first main result, Theorem 1.3, and Appendix D provides the missing details from our second main result, Theorem 1.4.

## A. Additional Related Work

Learning Gaussian Mixture Models (GMMs) is one of the most studied problems in statistics, dating back to Pearson (1894). Over the years, a plethora of works has explored this area (Dasgupta, 1999; Arora & Kannan, 2001; Vempala & Wang, 2002; Achlioptas & McSherry, 2005; Kannan et al., 2005; Brubaker & Vempala, 2008; Moitra & Valiant, 2010; Belkin & Sinha, 2015; Suresh et al., 2014; Daskalakis & Kamath, 2014; Hardt & Price, 2015; Diakonikolas et al., 2020; Bakshi et al., 2020; Diakonikolas et al., 2022; Liu & Moitra, 2021; Bakshi et al., 2022). Here, we provide a brief exposition of part of this literature, though this is not a comprehensive survey.

A key starting point was Dasgupta (1999), which studied learning GMMs with well-separated, spherical covariance components. Subsequent work by Vempala & Wang (2002); Achlioptas & McSherry (2005); Kannan et al. (2005) improved the separation condition to be dimension-independent. Hopkins & Li (2018) and Kothari et al. (2018) later refined the separation assumption to the information-theoretic limit, achieving this with quasi-polynomial-time algorithms. Notably, most of the aforementioned works extend beyond spherical Gaussians; however, they measure the pairwise mean separation between components relative to the largest eigenvalue of the components' covariance matrices. More recently, Liu & Li (2022) improved the runtime to polynomial for spherical Gaussians.

The case of arbitrary Gaussians with unknown component covariances has also been extensively studied (Belkin & Sinha, 2015; Moitra & Valiant, 2010; Bakshi & Kothari, 2020; Diakonikolas et al., 2020; Liu & Moitra, 2022; Bakshi et al., 2022). While the first works in this list had complexities doubly exponential in $k$, the number of components, the most recent have reduced this to $d^{O(k)}$. As noted in Section 1, hardness results from Diakonikolas et al. (2017); Bruna et al. (2021); Gupte et al. (2022) showed this complexity is necessary. However, certain special cases of GMMs can circumvent these lower bounds. This was the focus of Buhai & Steurer (2023) and Anderson et al. (2024), who studied GMMs with a minimum mixing weight $w_{\min} \geq 1/\operatorname{poly}(k)$ and unknown but shared covariance across components. These papers motivated us to study whether further improvements are possible for this family of GMMs.

## B. Additional Preliminaries

### B.1. Notation

We use $\mathbb{Z}_+$ to denote positive integers. For $n \in \mathbb{Z}_+$ we denote $[n] \overset{\text{def}}{=} \{1, \ldots, n\}$. We denote by $\mathbb{R}_0^+$ the set of all non-negative positive real numbers. We denote by $\exp(x) = e^x$ the exponential function, and by $\ln(x)$ the natural logarithm (the logarithm with base $e$). We write $x \otimes y$ to denote the tensor product between two vectors $x, y$. The tensor product of two vectors $u \in \mathbb{R}^m$ and $v \in \mathbb{R}^n$ is a matrix $u \otimes v \in \mathbb{R}^{m \times n}$ defined such that $(u \otimes v)_{ij} = u_i v_j$, where each entry is the product of the corresponding components of $u$ and $v$. This can be extended to product between more than two vectors. We denote $x^{\otimes k}$ the $k$-fold tensor product of the vector $x$ with itself. If $A$ is a tensor, $\|A\|_{\text{F}}$ denotes its Frobenius norm, which is the Euclidean norm of the vector obtained by stacking all entries of the tensor into a single vector. We write $x \sim \mathcal{D}$ for a random variable $x$ following the distribution $\mathcal{D}$ and use $\mathbf{E}[x]$ for its expectation. We use $\mathcal{N}(\mu, \Sigma)$ to denote the Gaussian distribution with mean $\mu$ and covariance matrix $\Sigma$. We write $\mathbf{Pr}(\mathcal{E})$ for the probability of an event $\mathcal{E}$. We denote by $\mathbf{1}(\mathcal{E})$ the indicator function of the event $\mathcal{E}$. The $L_p$ norm of a ($\mathbb{R}$-valued) random variable $x$ is defined to be $\|x\|_p = \mathbf{E}[|x|^p]^{1/p}$. The $L_p$ norm of a function $f : \mathbb{R}^d \to \mathbb{R}$ is defined to be the $L_p$ norm of the random variable $f(x)$, i.e., $\|f\|_p = \mathbf{E}_{x \sim \mathcal{N}(0,I)}[|f(x)|^p]^{1/p}$.

We use $a \lesssim b$ to denote that there exists an absolute universal constant $C > 0$ (independent of the variables or parameters on which $a$ and $b$ depend) such that $a \leq Cb$. Sometimes, we will also use the $O(\cdot), \Omega(\cdot), \Theta(\cdot)$ notation with the standard meaning.

### B.2. Hermite Analysis

Hermite polynomials form a complete orthogonal basis of the vector space $L^2(\mathbb{R}, \mathcal{N}(0,1))$ of all functions $f : \mathbb{R} \to \mathbb{R}$ such that $\mathbf{E}_{x \sim \mathcal{N}(0,1)}[f^2(x)] < \infty$. We will use the *normalized probabilist's* Hermite polynomials, which have unit norm

and are pairwise orthogonal with respect to the Gaussian measure, i.e., $\int_{\mathbb{R}} h_k(x) h_m(x) e^{-x^2/2} dx = \sqrt{2\pi} \mathbf{1}(k = m)$. These polynomials are the ones obtained by Gram-Schmidt orthonormalization of the basis $\{1, x, x^2, \ldots\}$ with respect to the inner product $\langle f, g \rangle_{\mathcal{N}(0,1)} := \mathbf{E}_{x \sim \mathcal{N}(0,1)}[f(x)g(x)]$. Every function $f \in L^2(\mathbb{R}, \mathcal{N}(0,1))$ can be uniquely written as $f(x) = \sum_{i=0}^{\infty} a_i h_i(x)$ and we have $\lim_{n \to \infty} \mathbf{E}_{x \sim \mathcal{N}(0,1)}[(f(x) - \sum_{i=0}^{n} a_i h_i(x))^2] = 0$ (see Andrews et al. (1999) for a more detailed exposition of Hermite polynomial's properties). We have the following closed form formula (see, e.g., Szegö (1989)):

$$h_n(x) = \sqrt{n!} \sum_{j=0}^{\lfloor n/2 \rfloor} \frac{(-1)^j}{j!(n-2j)!2^j} x^{n-2j} . \tag{10}$$

We will use the following fact, stating that the largest coefficient in the above expansion cannot be too large.

**Fact B.1** (Upper Bound on Hermite Polynomial Coefficients)**.** *Let $h_n(x)$ denote the normalized probabilist's Hermite polynomial of order $n$. In the monomial expansion $h_n(x) = \sum_{j=1}^{n} a_j x^j$, it holds $|a_j| \leq 2^{O(n)}$ for all $j \in [n]$.*

*Proof.* This follows by Equation (10). The $j$-th coefficient is $\left| \sqrt{n!}(-1)^j/(j!(n-2j)!2^j) \right| \leq \sqrt{n!}/(j!(n-2j)!2^j)$. Then one can use the elementary inequalities $e(k/e)^k \leq k! \leq ek(k/e)^k$ to bound the factorials that appear in the numerator and denominator and optimize the resulting function. The derivative analysis of the resulting function gives two points of zero derivative: $j = \frac{1}{4}(1 + 2n - \sqrt{1 + 4n})$ and $j = \frac{1}{4}(1 + 2n + \sqrt{1 + 4n})$. For each point, it can be checked that the function is smaller than $2^n$ in the limit $n \to \infty$. $\square$

**Definition B.2** (Ornstein-Uhlenbeck Operator)**.** For a $\rho \in [-1, 1]$, we define the *Ornstein-Uhlenbeck* (or *Gaussian noise*) operator $U_\rho$ as the operator that maps a distribution $F$ on $\mathbb{R}$ to the distribution of the random variable $\rho X + \sqrt{1 - \rho^2} Z$, where $X \sim F$ and $Z \sim \mathcal{N}(0, 1)$ independently of $X$.

A well-known property of *Ornstein–Uhlenbeck* operator is that it operates diagonally with respect to Hermite polynomials.

**Fact B.3** (see, e.g., O'Donnell (2014))**.** *For any normalized Hermite polynomial $h_i$, any distribution $F$ on $\mathbb{R}$, and $\delta \in [-1, 1]$, it holds that $\mathbf{E}_{X \sim U_\rho F}[h_i(X)] = \rho^i \mathbf{E}_{X \sim F}[h_i(X)]$.*

### B.3. Properties of Polynomials Under the Gaussian Measure

**Fact 2.1** (Gaussian Moments)**.** $\mathbf{E}_{x \sim \mathcal{N}(0,1)}[x^t] \lesssim (t/e)^{t/2} \; \forall t \geq 0$.

**Fact B.4** (Carbery-Wright Inequality (Carbery & Wright, 2001))**.** *There is an absolute constant $C$ such that the following holds. Let $q, \gamma \in \mathbb{R}_0^+$, $\mu \in \mathbb{R}^d$, $\Sigma \in \mathbb{R}^{d \times d}$ such that $\Sigma$ is symmetric PSD and $p : \mathbb{R}^d \to \mathbb{R}$ be a multivariate polynomial of degree at most $r$. Then*

$$\mathbf{Pr}_{x \sim \mathcal{N}(\mu, \Sigma)} (|p(x)| \leq \gamma) \leq \frac{Cq\gamma^{1/r}}{\left( \mathbf{E}_{z \sim \mathcal{N}(\mu, \Sigma)} \left[ |p(z)|^{q/r} \right] \right)^{1/q}} .$$

The way that we will apply this is with the following choice of parameters: $d = 1, \mu = 0, \Sigma = 1, q = r$ and $\gamma = \epsilon \|p\|_1$, where $\epsilon$ is a new parameter. This gives:

**Fact 2.2.** *For every polynomial of degree $r$ and every $\epsilon > 0$, $\mathbf{Pr}_{x \sim \mathcal{N}(0,1)} (|p(x)| \leq \epsilon \|p\|_1) \leq O(r \epsilon^{1/r})$.*

The following inequality can be easily derived using Hölder's inequality.

**Fact B.5.** $\|x\|_2 \leq \|x\|_1^{1/3} \|x\|_4^{2/3}$ *for any random variable.*

The following inequality is the Gaussian Hypercontractivity property (see, e.g., Bogachev (1998); Nelson (1973))

**Fact 2.3** (Gaussian Hypercontractivity)**.** *If $p$ is a degree $r$ polynomial and $k > 2$, then $\|p\|_k \leq (k-1)^{r/2} \|p\|_2$.*

In particular we will use the above in the following way:

**Fact 2.4.** *For any polynomial $p$ of degree $r$, $\frac{\|p\|_1}{\|p\|_2} \geq 3^{-r}$.*

*Proof.* $\|p\|_1 \geq \frac{\|p\|_2^3}{\|p\|_4^2} = \|p\|_2 \left(\frac{\|p\|_2}{\|p\|_4}\right)^2 \geq 3^{-t}\|p\|_2$, where the first step used Fact B.5 and the last step used Gaussian Hypercontractivtiy (Fact 2.3) with $k = 4$. Rearranging completes the proof. $\qquad\square$

**Fact B.6** (Gaussian Norm Concentration). *If $x \sim \mathcal{N}(\mu, I)$, with probability $1 - \tau$ we have that*

$$\left|\|x\|^2 - \left(\|\mu\|^2 + d\right)\right| \lesssim \log \frac{1}{\tau} + \left(\sqrt{d} + \|\mu\|\right)\sqrt{\log \frac{1}{\tau}}.$$

### B.4. Arithmetic Mean-Geometric Mean Inequality

In this paper, we will use a continuous analog of the *Arithmetic Mean-Geometric Mean* (AM-GM) inequality. The continuous analog for the arithmetic mean of a sequence $\frac{1}{n}\sum_{i=1}^n x_i$ is what one obtains by replacing the summation with its continuous counterpart. Specifically, the arithmetic mean of a function $f : \mathbb{R} \to \mathbb{R}$ over an interval $I$ is defined as: $(1/|I|) \int_I f(x)\mathrm{d}x$.

The geometric mean of a discrete sequence is $\prod_{i=1}^n x_i^{1/n}$. Its generalization relies on the property that $\ln \prod_{i=1}^n x_i^{1/n} = \frac{1}{n}\sum_{i=1}^n \ln x_i$ (assuming $x_i > 0$). By replacing the summation with an integral, the generalization of the geometric mean of a function $f$ over an interval $I$ is: $\exp\left(\frac{1}{|I|}\int_I \ln f(x)\mathrm{d}x\right)$. The continuous analog of the AM-GM inequality thus is the following statement. The proof follows directly from Jensen's inequality:

**Fact 2.5** (Continuous AM-GM Inequality). *Let $f : \mathbb{R} \to \mathbb{R}_0^+$ be a function, and let $I \subseteq \mathbb{R}$ be a finite interval. If $f(x)$ and $\ln f(x)$ are integrable on $I$, then the following holds: $\frac{1}{|I|}\int_I f(x)\mathrm{d}x \geq \exp\left(\frac{1}{|I|}\int_I \ln f(x)\mathrm{d}x\right)$.*

### B.5. Statistical Query Lower Bounds Background

We first restate the definition of the non-Gaussian component analysis (NGCA) hypothesis testing problem.

**Problem 2.6** (Non-Gaussian Component Analysis (NGCA)). Let $B$ be a distribution on $\mathbb{R}$. For a unit vector $v$, we denote by $P_{B,v}$ the distribution with the density $P_{B,v}(x) := B(v^\top x)\phi_{\perp v}(x)$, where $\phi_{\perp v}(x) = \exp\left(-\|x - (v^\top x)v\|_2^2/2\right)/(2\pi)^{(d-1)/2}$, i.e., the distribution that coincides with $B$ on the direction $v$ and is standard Gaussian in every orthogonal direction. We define the following hypothesis testing problem:

- $H_0$: The data distribution is $\mathcal{N}(0, I_d)$.
- $H_1$: The data distribution is $P_{B,v}$, for some vector $v \in \mathcal{S}^{d-1}$ in the unit sphere.

**Condition B.7** (Approximate moment matching). Let $m \in \mathbb{Z}_+$. The distribution $B$ on $\mathbb{R}$ is such that $\mathbf{E}_{x \sim B}[x^i] - \mathbf{E}_{x \sim \mathcal{N}(0,1)}[x^i]| \leq \nu$ for all $i \in [m]$.

A known result is that the NGCA problem of Problem 2.6 is hard in the SQ model if $B$ matches a lot of moments with the standard Gaussian. This was shown in (Diakonikolas et al., 2017) and was later strengthened in Diakonikolas et al. (2023). The following is Theorem 1.5 in Diakonikolas et al. (2023) using $\lambda = 1/2$ and $c = (1 - \lambda)/8 = 1/16$.

**Proposition B.8** (Theorem 1.5 in Diakonikolas et al. (2023)). *Let $d, m$ be positive integers with $d \geq (m \log d)^2$. Any SQ algorithm that solves Problem 2.6 for a distribution $B$ satisfying Condition B.7 requires either $2^{d^{\Omega(1)}}$ many queries or at least one query with accuracy $d^{-m/16} + (1 + o(1))\nu$.*

## C. Omitted Details from Section 3

We restate and prove Theorem 1.3.

**Theorem 1.3** (SQ Lower Bound for Uniform Weights). *Let $C$ be a sufficiently large absolute constant, $k > C$ and $d \geq (\log k \log d)^2$ be integers. If we further restrict the alternative hypothesis in Problem 1.1 to have $w_i = 1/k$ for all $i \in [k]$, any SQ algorithm requires either $2^{d^{\Omega(1)}}$ queries or at least one query of accuracy $d^{-\Omega(\log k)}$.*

*Proof.* Let $S$ be the set from Proposition 4.2 and $A$ be the uniform distribution on $S$. That is, $A$ is a discrete distribution supported on $k$ points and is guaranteed to match the first $m = \Omega(\log k)$ moments with $\mathcal{N}(0, 1)$. Let $B = U_\rho A$ be the distribution which is obtained by applying the Ornstein-Uhlenbeck operator (Definition B.2) with $\rho = \sqrt{\delta}$. Then $B$ is a

$k$-GMM with uniform weights and variance $1 - \delta$ for each component. Moreover, for every $t = 0, 1, \ldots, m$ we have the following for the $i$-th Hermite polynomial

$$\mathop{\mathbf{E}}_{x \sim B}[h_i(x)] = \mathop{\mathbf{E}}_{x \sim U_\rho A}[h_i(x)] = \rho^i \mathop{\mathbf{E}}_{x \sim A}[h_i(x)] = \rho^i \mathop{\mathbf{E}}_{x \sim \mathcal{N}(0,1)}[h_i(x)] \,, \tag{11}$$

where the above uses Fact B.3 and the moment matching property of $A$. Since $\mathbf{E}_{x \sim \mathcal{N}(0,1)}[h_i(x)] = 1$ for $i = 0$ and $\mathbf{E}_{x \sim \mathcal{N}(0,1)}[h_i(x)] = 0$ for all $i > 0$, Equation (11) means that $\mathbf{E}_{x \sim B}[h_i(x)] = \mathbf{E}_{x \sim \mathcal{N}(0,1)}[h_i(x)]$, i.e., $B$ matches the first $m$ moments with $\mathcal{N}(0, 1)$.

An application of Proposition B.8 with $\nu = 0$ shows that the NGCA Problem 2.6 that uses the distribution $B$ from above has SQ complexity $d^{\Omega(\log k)}$. Noting that this problem is equivalent to Problem 1.1 completes the proof of Theorem 1.3.

We conclude by addressing an edge case. The proof above implicitly assumes that the set $S$ contains distinct points (as otherwise, the weights in the corresponding GMM might not all be exactly $1/k$). Here, we argue that Theorem 1.3 still holds even if $S$ contains duplicates. Specifically, one can perturb each point in $S$ by a at most an arbitrarily small amount $\Delta$, ensuring that the points become distinct and that the moments in the resulting GMM distribution $B$ are being matched up to an error of $\nu$ rather than exactly (note that for any $\nu$ we can find a perturbation so that the moment gap is no more than $\nu$). The SQ lower bound from Proposition B.8 then implies that we either require $2^{d^{\Omega(1)}}$ queries or at least one query with accuracy $d^{-m/16} + (1 + o(1))\nu$. By choosing $\Delta$ appropriately small, we can ensure that $\nu < d^{-m/16}$. $\qquad\square$

# D. Omitted Details from Section 4

## D.1. Omitted Details from Section 4.1

The lemma below shows that if the approximate momement matching condition is violated, then it has to be violated by a monomial (up to a small deterioration of parameters).

**Lemma D.1.** *Let $C$ be a sufficiently large absolute constant. If there exists a polynomial $g : \mathbb{R} \to \mathbb{R}$ of degree $r$ and unit norm ($\mathbf{E}_{x \sim \mathcal{N}(0,1)}[g^2(x)] = 1$) such that*

$$\left| \mathop{\mathbf{E}}_{x \sim A}[g(x)] - \mathop{\mathbf{E}}_{x \sim \mathcal{N}(0,1)}[g(x)] \right| > \gamma \,,$$

*then there exists a monomial $x^i$ with $i \leq r$ for which*

$$\left| \mathop{\mathbf{E}}_{x \sim A}[x^i] - \mathop{\mathbf{E}}_{x \sim \mathcal{N}(0,1)}[x^i] \right| > 2^{-C \cdot r}\gamma \,.$$

*Proof.* We will show this by contradiction. Suppose that every monomial of degree $i \leq r$ satisfies $\left| \mathbf{E}_{x \sim A}[x^i] - \mathbf{E}_{x \sim \mathcal{N}(0,1)}[x^i] \right| \leq 2^{-Cr}\gamma$. Then, if we expand $g(x)$ in the hermite basis, i.e., $g(x) = \sum_{i=1}^{r} a_i h_i(x)$, we have

$$\left| \mathop{\mathbf{E}}_{x \sim A}[g(x)] - \mathop{\mathbf{E}}_{x \sim \mathcal{N}(0,1)}[g(x)] \right| \leq \sum_{i=1}^{r} |a_i| \left| \mathop{\mathbf{E}}_{x \sim A}[h_i(x)] - \mathop{\mathbf{E}}_{x \sim \mathcal{N}(0,1)}[h_i(x)] \right|$$

$$\leq \sqrt{\sum_{i=1}^{r} |a_i|^2} \sqrt{\sum_{i=1}^{r} \left| \mathop{\mathbf{E}}_{x \sim A}[h_i(x)] - \mathop{\mathbf{E}}_{x \sim \mathcal{N}(0,1)}[h_i(x)] \right|^2} \,, \tag{12}$$

where the second step uses Cauchy-Schwarz inequality. To further upper bound this, first note that $\sqrt{\sum_{i=1}^{r} |a_i|^2} = \|g\|_2 = 1$, by Parseval's identity and our assumption of unit norm. For the other factor above, we can write $h_i(x) = \sum_{j=1}^{i} b_{ij}x^j$ and use the fact that $|b_{ij}| \leq 2^{O(i)}$ (Fact B.1). Then,

$$\left| \mathop{\mathbf{E}}_{x \sim A}[h_i(x)] - \mathop{\mathbf{E}}_{\mathcal{N}(0,1)}[h_i(x)] \right| \leq \sum_{j=1}^{i} |b_{ij}| \left| \mathop{\mathbf{E}}_{x \sim A}[x^i] - \mathop{\mathbf{E}}_{\mathcal{N}(0,1)}[x^i] \right|$$

$$\leq 2^{O(r)} \sum_{j=1}^{i} \left| \mathop{\mathbf{E}}_{x \sim A}[x^i] - \mathop{\mathbf{E}}_{\mathcal{N}(0,1)}[x^i] \right| \leq 2^{O(r)} r 2^{-Cr}\gamma < 2^{-Cr/2}\gamma \,,$$

Plugging that in Equation (12), we obtain $|\mathbf{E}_{x\sim A}[g(x)] - \mathbf{E}_{\mathcal{N}(0,1)}[g(x)]| \leq \sqrt{r}2^{-Cr/2}\gamma < \gamma$, which gives the desired contradiction. $\square$

The lemma below provides a testing algorithm for the NGCA problem (Problem 2.6) in the special case where the distribution $B$ is $k$-GMM for which a moment of order at most $\widetilde{m}$ is guaranteed to be significantly different than the corresponding moment of $\mathcal{N}(0,1)$.

**Lemma D.2** (Testing Algorithm for Parallel Pancakes when the $m$-th Moment Deviates). *Let $B$ be a Gaussian mixture on $\mathbb{R}$ of the form $B = \sum_{i=1}^{k} w_i \mathcal{N}(\mu_i, \sigma^2)$, where $\sigma \in (0,1)$ and $w_i \geq w_{\min}$ for all $i \in [k]$. For a decreasing sequence $\lambda_m$, denote by $m$ the biggest integer such that every degree-$m' \leq m$ polynomial $g$ satisfies*

$$\left| \mathbf{E}_{x\sim B}[g(x)] - \mathbf{E}_{x\sim\mathcal{N}(0,1)}[g(x)] \right| \leq \lambda_m \sqrt{\mathbf{E}_{x\sim\mathcal{N}(0,1)}[g^2(x)]} \,, \tag{13}$$

*Consider the non-Gaussian component analysis hypothesis testing Problem 2.6. Let $\widetilde{m}$ be any upper bound for $m$ i.e., $m \leq \widetilde{m}$. There is an algorithm that takes as input $\widetilde{m}$ and $w_{\min}$, draws $n = \left( (\widetilde{m}d)^{O(\widetilde{m})}\lambda_{\widetilde{m}}^{-O(1)} + \log(k)w_{\min}^{-1} \right)\log(1/\tau)$ samples, and distinguishes correctly between $H_0$ and $H_1$ with probability $1 - \tau$. Moreover, the runtime of the algorithm is polynomial in $n$ and $d$.*

*Proof.* We will do the proof for $\widetilde{m} = m$. The proof trivially extends to any $\widetilde{m}$ bigger than $m$. For degree $m + 1$, there exists a polynomial $g$ that violates the condition in Equation (13). By Lemma D.1, there exists a monomial $x^i$ with $i \leq m + 1$ such that

$$\widetilde{\lambda} = \left| \mathbf{E}_{x\sim B}[x^i] - \mathbf{E}_{\mathcal{N}(0,1)}[x^i] \right| > 2^{-C\cdot m}\lambda_m \,.$$

For the corresponding $d$-dimensional distributions $P_{B,v}$ (defined in Problem 2.6) and $\mathcal{N}(0,I)$, we have

$$\mathbf{E}_{x\sim P_{B,v}}[x^{\otimes i}] - \mathbf{E}_{x\sim\mathcal{N}(0,I)}[x^{\otimes i}] = \pm\widetilde{\lambda}v^{\otimes i} \,.$$

Thus, the Frobenius norm is

$$\left\| \mathbf{E}_{x\sim P_{B,v}}[x^{\otimes i}] - \mathbf{E}_{x\sim\mathcal{N}(0,I)}[x^{\otimes i}] \right\|_{\mathrm{F}} = \widetilde{\lambda} > 2^{-C\cdot m}\lambda_m \,.$$

This means that at least one entry in the difference of the two tensors has gap at least $\epsilon := d^{-(m+1)}2^{-C\cdot m}\lambda_m$. The idea for the testing algorithm is to approximate every entry of $\mathbf{E}_{x\sim P_{B,v}}[x^{\otimes i}] - \mathbf{E}_{x\sim\mathcal{N}(0,I)}[x^{\otimes i}]$ up to absolute error $\epsilon/100$, and test whether some entry is bigger than $\epsilon/2$. This is done in Algorithm 2 and Algorithm 3.

---

**Algorithm 2** Testing Algorithm

1: **Input**: $k, \widetilde{m} \in \mathbb{Z}_+$, $w_{\min} \in (0,1]$.
2: **Output**: $\hat{H} \in \{H_0, H_1\}$.

3: **for** $i = 1, 2, 3, \ldots, \widetilde{m} + 1$ **do**
4:     Run Algorithm 3 with input $k, i, \widetilde{m}, w_{\min}$ repetitively $\log((\widetilde{m}+1)/\tau)$ times and let $\hat{H}$ be the most frequent output.
5:     **if** $\hat{H} = H_1$ **then**
6:         Return $H_1$
7: Return $H_0$.

---

We start with the correctness of the sub-routine, Algorithm 3. We say that that the output of Algorithm 3 is "successful" if it always agrees with the true hypothesis, with the exception of the following case, where mistakes are permitted: this case is when the true hypothesis is $H_1$, the data distribution satisfies $\max_{i\in[k]}\|\mu_i\|_2 \leq C\sqrt{d}$ (recall that $\mu_i$'s are the centers of the $k$-GMM distribution $B$) and $\left\| \mathbf{E}_{x\sim P_{B,v}}[x^{\otimes i}] - \mathbf{E}_{x\sim\mathcal{N}(0,I)}[x^{\otimes i}] \right\|_{\mathrm{F}} \leq 2^{-Cm}\lambda_m$. We will show that the output of Algorithm 3 is indeed "successful" in this sense with constant probability.

---

**Algorithm 3** Checking the $i$-th order tensor mismatch

---

1: **Input**: $k, \widetilde{m} \in \mathbb{Z}_+, i \in \mathbb{Z}_+, w_{\min} \in (0, 1]$.
2: **Output**: $\hat{H} \in \{H_0, H_1\}$.

3: Define $n = (\widetilde{m}d)^{C\widetilde{m}} \lambda_{\widetilde{m}}^{-C} + \log(k) w_{\min}^{-1}$ for sufficiently large $C$.
4: Draw $x_1, \ldots, x_n$ i.i.d. from the data distribution.
5: **if** there exists $i \in [n]$ with $\|x_i\|_2 > C\sqrt{d}$ **then**
6:     Output $H_1$ and terminate.
7: **else**
8:     Form the empirical tensor $M = \mathbf{E}_{x \sim S}[x^{\otimes i}]$.
9:     Let $M'$ denote the Gaussian tensor $\mathbf{E}_{x \sim \mathcal{N}(0,I)}[x^{\otimes i}]$.
10:     **if** there is an entry in $M_{i_1,\ldots,j_i}$ with $|M_{i_1,\ldots,j_i} - M'_{i_1,\ldots,j_i}| > d^{-C\widetilde{m}} \lambda_{\widetilde{m}}$ **then**
11:         Output $H_1$ and terminate.
12: Return $H_0$.

---

Having that claim established, Lemma D.2 follows straightforwardly: First, note that the probability of success can be amplified to $1 - \tau$ by repeating the subroutine $\log(1/\tau)$ times and taking the majority vote. Second, if the true hypothesis is $H_1$, there exists an $i$ such that $\left\|\mathbf{E}_{x \sim P_{B,v}}[x^{\otimes i}] - \mathbf{E}_{x \sim \mathcal{N}(0,I)}[x^{\otimes i}]\right\|_{\mathrm{F}} > 2^{-Cm} \lambda_m$. Combined with the claim of the previous paragraph about Algorithm 3, this ensures that running Algorithm 3 for that $i$ will be $H_1$, as desired. Similarly, under $H_0$, the output is always $H_0$, which guarantees that the output of Algorithm 2 matches the true hypothesis.

We now move to showing the claim that Algorithm 3 is "successful" with constant probability. We examine the following cases:

**Case 1** The true hypothesis is $H_0$. In this case, the data distribution is $D = \mathcal{N}(0, I)$. By Gaussian norm concentration (Fact B.6) we have $\mathbf{Pr}_{x_1,\ldots,x_n \sim \mathcal{N}(0,I)}[\max_i \|x_i\| > C\sqrt{d \log n}] < 0.01$. This means that Algorithm 3 will enter line 7. Then, by standard entry-wise concentration of Gaussian tensors (see e.g., Fact 5.6 and Equation (5.4) in (Kothari & Steurer, 2017)) we have that if $n > d^{C'm}/\lambda_m^2$ for $C' \gg C$, we will have $\|\mathbf{E}_{x \sim \mathcal{N}(0,I)}[x^{\otimes i}] - \mathbf{E}_{x \sim S}[x^{\otimes i}]\|_\infty < d^{-Cm} \lambda_m$ and thus the condition in 10 will be false, resulting in the algorithm to output $H_0$.

**Case 2** The hypothesis under effect is $H_1$ and $\max_{i \in [k]} \|\mu_i\|_2 > C\sqrt{d \log n}$. The claim is that for $\log(k)/w_{\min}$ samples, with high constant probability, one sample from every component will be observed, and the sample that comes from the component with $\|\mu_i\|_2 > C\sqrt{d \log n}$ will satisfy $\|x\| > C\sqrt{d \log n}$ by Fact B.6. To see the first part of the claim, fix an $i \in [k]$. With $10/w_{\min}$ samples, one sample from $i$ will be observed with at least $0.9$ probability. We can boost that probability to $1 - 1/k$ by repeating $\log(k)$ times. Then, by union bound, this means that one sample from each component is observed with constant probability.

**Case 3** The hypothesis under effect is $H_1$ and $\max_{i \in [k]} \|\mu_i\|_2 \leq C\sqrt{d \log n}$. In this case the data distribution is a $k$-GMM where the center of every Gaussian component is bounded in norm by most $R = C\sqrt{d \log n}$. By Gaussian norm concentration, if $x_1, \ldots, x_n$ are points drawn from that GMM, then with constant probability we will have $\|x_i\|_2 \leq 2C\sqrt{d \log n}$. Therefore the algorithm will enter 7, and because of the bound $\|x_i\|_2 \leq 2C\sqrt{d \log n}$, we can treat the distribution as bounded and use Hoeffding bound for the tensor concentration. That application of Hoeffding's inequality shows that if $n > R^{C'm} d^{C'm}/\lambda_m^{C'}$ then the estimation error is at most $d^{-Cm} \lambda_m$. Thus, in this case, the algorithm will output $H_1$ if and only if $\left\|\mathbf{E}_{x \sim A}[x^{\otimes i}] - \mathbf{E}_{x \sim \mathcal{N}(0,1)}[x^{\otimes i}]\right\|_{\mathrm{F}} \leq 2^{-Cm} \lambda_m$.

This completes the proof of the claim. $\qquad\square$

Our main result, Theorem 1.4 will be based on Lemma D.2 and our impossibility of matching result, Proposition 4.2 that will allow us to use $\widetilde{m} = O(\log(k) + k')$ in Lemma D.2. However, Proposition 4.2 concerns only discrete distributions, while the parallel pancakes uses a Gaussian mixture. In order to bridge this difference, we show the following lemma, which states that the impossibility of moment matching can be indeed extended to Gaussian mixtures.

**Lemma D.3.** *Let $P$ be a Gaussian mixture distribution on $\mathbb{R}$ of the form $B = \sum_{i=1}^k w_i \mathcal{N}(\mu_i, 1 - \delta)$, where $w_i > 0$ with $\sum_{i=1}^k w_i = 1$ are the weights of each component, $\mu_1, \ldots, \mu_k \in \mathbb{R}$ are the centers and $\delta \in (0, 1]$ is the parameter*

*associated with the common variance of the components. Suppose that for every polynomial of degree at most $m'$ and* $\mathbf{E}_{x \sim \mathcal{N}(0,1)}[p^2(x)] = 1$ *the following holds*

$$\left| \mathbf{E}_{x \sim B}[p(x)] - \mathbf{E}_{x \sim \mathcal{N}(0,1)}[p(x)] \right| \leq \lambda \ . \tag{14}$$

*Then, if $A$ denotes the discrete distribution on $\{\mu_1/\sqrt{\delta}, \dots, \mu_i/\sqrt{\delta}\}$ that assigns mass $w_i$ to the point $\mu_i/\sqrt{\delta}$ for $i \in [k]$, the following is true: For every polynomial with degree at most $m'$ and $\mathbf{E}_{x \sim \mathcal{N}(0,1)}[p^2(x)] = 1$ it holds*

$$\left| \mathbf{E}_{x \sim A}[p(x)] - \mathbf{E}_{x \sim \mathcal{N}(0,1)}[p(x)] \right| \leq \sqrt{m'}\, \lambda\, \delta^{-m'/2} \ . \tag{15}$$

*Proof.* We can write the distribution $B$ as the result of applying the Ornstein-Uhlenbeck (Definition B.2) operator to $A$, i.e., $B = U_\rho A$ with $\rho = \sqrt{\delta}$. By Fact B.3, we have the following for every $i = 1, 2, \dots$:

$$\mathbf{E}_{x \sim A}[h_i(x)] = \delta^{-i/2} \mathbf{E}_{x \sim B}[h_i(x)] \ . \tag{16}$$

Fix $i \in [m']$. Using the above and the fact that $B$ matches approximately the $m$ first moments with $\mathcal{N}(0,1)$ (in the sense of Equation (14)) we have the following for the gap between the expectations of the Hermite polynomial $h_i$ under $A$ and $\mathcal{N}(0,1)$:

$$
\begin{aligned}
\left| \mathbf{E}_{x \sim A}[h_i(x)] - \mathbf{E}_{x \sim \mathcal{N}(0,1)}[h_i(x)] \right| &= \left| \delta^{-i/2} \mathbf{E}_{x \sim B}[h_i(x)] - \mathbf{E}_{x \sim \mathcal{N}(0,1)}[h_i(x)] \right| && \text{(using Equation (16))} \\
&= \left| \delta^{-i/2} \mathbf{E}_{x \sim B}[h_i(x)] \right| && \text{(using } \mathbf{E}_{x \sim \mathcal{N}(0,1)}[h_i(x)] = 0 \text{ for } i \geq 1) \\
&\leq \delta^{-i/2} \left( \left| \mathbf{E}_{x \sim \mathcal{N}(0,1)}[h_i(x)] \right| + \lambda \right) && \text{(using Equation (14))} \\
&= \delta^{-i/2} \lambda && \text{(using } \mathbf{E}_{x \sim \mathcal{N}(0,1)}[h_i(x)] = 0 \text{ for } i \geq 1)
\end{aligned}
$$

For the special case $i = 0$, we have exact matching, $\mathbf{E}_{x \sim A}[h_0(x)] = \mathbf{E}_{x \sim \mathcal{N}(0,1)}[h_0(x)]$ since $h_0(x) = 1$.

Now, in order to show Equation (15), consider a general polynomial $p(x)$ with $\mathbf{E}_{x \sim \mathcal{N}(0,1)}[p^2(x)] = 1$. Expanding in the Hermite basis, we can write $p(x) = \sum_{i \in [m']} a_i h_i(x)$ with $\sum_{i \in [m']} a_i^2 = 1$ (which means that $\mathbf{E}_{x \sim \mathcal{N}(0,1)}[p^2(x)] = 1$ by Parseval's identity). We have

$$
\begin{aligned}
\left| \mathbf{E}_{x \sim A}[p(x)] - \mathbf{E}_{x \sim \mathcal{N}(0,1)}[p(x)] \right| &\leq \sqrt{\sum_{i=1}^{m'} a_i^2} \sqrt{\sum_{i=1}^{m'} \left| \mathbf{E}_{x \sim A}[h_i(x)] - \mathbf{E}_{x \sim \mathcal{N}(0,1)}[h_i(x)] \right|^2} \\
&\leq \sqrt{m'} \max_{i \in [m']} \left| \mathbf{E}_{x \sim A}[h_i(x)] - \mathbf{E}_{x \sim \mathcal{N}(0,1)}[h_i(x)] \right| \\
&\leq \sqrt{m'}\, \delta^{-m'/2} \lambda \ .
\end{aligned}
$$

$\square$

We now combine the previous statements to show our main theorem.

**Theorem 1.4** (Testing Algorithm for Parallel Pancakes). *Consider the version of the parallel pancakes hypothesis testing problem (Problem 1.1), where $k' \leq k$ of the weights $w_i$ in the Gaussian mixture are unconstrained and the remaining $k - k'$ are assumed to be equal to each other. There is an algorithm for that problem which draws $n = O\left( (kd/\delta)^{O(k' + \log(k))} + \log(k)/w_{\min} \right)$ samples (where $\delta$ is as in Problem 1.1 and $w_{\min} = \min_{i \in [k]} w_i$ is the smallest weight), has runtime polynomial in $n, d$, and it outputs the correct hypothesis with probability at least $0.99$.*

*Proof.* First, we note that the parallel pancakes testing problem of interest is a special case of the non-Gaussian component analysis Problem 2.6 where $B = \sum_{i \in [k]} w_i \mathcal{N}(\mu_i, 1 - \delta)$, where $w_i, \mu_i$ and $\delta$ the ones from Problem 1.1, in particular $k'$ of the $w_i$'s are unconstrained and the rest are assumed to be uniform.

The proof consists of two parts: The first part argues that this one-dimensional distribution $B$ cannot match approximately more than the first $m = O(\log k + k')$ moments with $\mathcal{N}(0, I)$ (the approximate moment matching will be quantified shortly). Then, for the second part, we can show that since the $m + 1$ moment deviates significantly from that of $\mathcal{N}(0, I)$, estimating the empirical moment tensor of order $m + 1$ and comparing with the one from $\mathcal{N}(0, I)$ yields a successful test.

We now proceed with the quantification. Let $C$ be a sufficiently large constant, and define $m$ to be the largest integer such that for every polynomial $p$ of degree $m' \leq m$ and $\|p\|_2 = 1$ we have

$$\left| \mathop{\mathbf{E}}_{x \sim A}[p(x)] - \mathop{\mathbf{E}}_{x \sim \mathcal{N}(0,1)}[p(x)] \right| \leq \lambda_m . \tag{17}$$

To prove our claim by contradiction, suppose that $m > C(k' + \log k)$. For each degree $m' \leq m$ We will use Lemma D.3 with $\lambda = (\delta/2)^{Cm}$. The application of Lemma D.3 yields that the discrete distribution $A$ supported on a scaled version of the centers $\mu_i$ and using the same weights $w_i$ approximately matches the same $m$ first moments with $\mathcal{N}(0, 1)$, i.e., for every polynomial $p$ of degree $m' \leq m$ and $\|p\|_2 = 1$ we have

$$\left| \mathop{\mathbf{E}}_{x \sim D}[p(x)] - \mathop{\mathbf{E}}_{x \sim \mathcal{N}(0,1)}[p(x)] \right| \leq \sqrt{m} \lambda_m \delta^{-m/2} \leq 2^{-Cm/2} . \tag{18}$$

where the last step uses that $\lambda = (2\delta)^{-Cm}$.

The conclusion of Equation (18) contradicts Proposition 4.1. This is because the discrete distribution $D$ from above, is of the form that Proposition 4.1 considers: supported on $k$ points, with $k'$ of the points having arbitrary mass and the remaining $k - k'$ having equal masses.

Thus far, we have shown that if $m$ is the largest degree for which all moments $m' \leq m$ of the distribution $A$ match with $\mathcal{N}(0, 1)$ in the sense of Equation (17), then $m = O(\log(k) + k')$.

The result then follows by Lemma D.2 with $\widetilde{m} = C(\log(k) + k')$ for a sufficiently large constant $C$, $\sigma^2 = 1 - \delta$, and $\lambda_{\widetilde{m}} = (\delta/2)^{C\widetilde{m}}$.

$\square$

### D.2. Omitted Details from Section 4.2

We restate and prove a version of Proposition 4.2 which does not involve minimum weight $w_0$ of points with equal weights.

**Proposition 4.1.** *Let $k' < k$ be positive integers, and let $A$ be a discrete distribution on $k$ points in $\mathbb{R}$. Suppose $k - k'$ of the points have equal probability masses, while the remaining $k'$ points have unrestricted probability masses. Denote by $m$ the highest degree for which every degree-$m' \leq m$ polynomial $g$ satisfies $\left| \mathbf{E}_{x \sim A}[g(x)] - \mathbf{E}_{x \sim \mathcal{N}(0,1)}[g(x)] \right| \leq 2^{-C \cdot m} \|g\|_2$, then $m$ must satisfy $m \leq O(\log k) + O(k')$.*

*Proof.* Suppose that the order $m$ is bigger than $C \log k + Ck'$. If $C$ is sufficiently large, we will show that this moment matching is impossible.

Let $\mu_1, \ldots, \mu_k$ be the points on which $A$ is supported, and by $w_1, \ldots, w_k$ the probability masses of the points. Without loss of generality, assume that the first $k'$ points are the ones which do not have any restriction on their probability mass, and the remaining $k - k'$ are the points with equal probability masses ($w_i = w_j$ for all $i, j \in \{k' + 1, \ldots, k\}$ with $i \neq j$). Let $p(x) = (x - \mu_1) \cdots (x - \mu_{k'})$ be the polynomial whose roots are the first $k'$ points.

We will show the following series of inequalities (we use the notation $\|p\|_r = \mathbf{E}_{x \sim \mathcal{N}(0,1)}[|p(x)|^r]^{1/r}$):

$$\sum_{i=k'+1}^{k} w_i \geq \left( \frac{\mathbf{E}_{x \sim A}[p(x)]}{\mathbf{E}_{x \sim A}[p^2(x)]^{1/2}} \right)^2 \gtrsim \left( \frac{\|p\|_1}{\|p\|_2} \right)^2 \geq 3^{-2k'} . \tag{19}$$

The third step is Fact 2.4. To see how the first step is derived, let $\mathcal{E}$ be the event that $i \in \{k'+1, \dots, k\}$. Then

$$\|p\|_1 = \mathop{\mathbf{E}}_{\mu_i \sim A}[p(\mu_i)] = \mathop{\mathbf{E}}_{\mu_i \sim A}[p(\mu_i)\mathbb{1}(\mathcal{E})] \leq \sqrt{\mathop{\mathbf{E}}_{\mu_i \sim A}[p^2(\mu_i)]}\sqrt{\mathbf{Pr}[\mathcal{E}]} = \|p\|_2\sqrt{\mathbf{Pr}[\mathcal{E}]} = \|p\|_2\sqrt{\sum_{i=k'+1}^{k} w_i} \ ,$$

where the first step above uses the fact that $\mu_1, \dots, \mu_{k'}$ are roots of $p$. Rearranging gives $\sum_{i=k'}^{k} w_i \geq (\|p\|_1/\|p\|_2)^2$.

It remains to show the second step in Equation (19), which is due to the approximate moment matching property: Let $\lambda_m := 2^{-Cm}$ to save space. For the numerator, we have $\mathbf{E}_{x \sim A}[p(x)] \geq \|p\|_1 - \lambda_m\|p\|_2 \geq \|p\|_1(1 - \lambda_m 3^{k'}) \geq \|p\|_1/2$, where the first step uses the approximate moment matching, the second step uses Fact 2.4 and the last part uses that $\lambda_m := w_0 2^{-Cm}$ with $C$ being sufficiently large constant and $m > k'$. We can work similarly for the denominator to get $\mathbf{E}_{x \sim A}[p^2(x)] \leq \|p\|_2^2 + \lambda_m\|p\|_4^2 \leq \|p\|_2^2(1 + \lambda_m 3^{k'}) \leq 2\|p\|_2^2$, where we used Fact 2.3 in the penultimate step. Combining the bounds for numerator and denominator conclude the proof of the second step in Equation (19).

We can now conclude the proof of Theorem 1.4. Since we have assumed that the weights for the last $k - k'$ points are equal to each other, Equation (19) implies that $\min_{i=k'+1,\dots,k} w_i \geq 3^{-2k'}/k$. Using Proposition 4.2 with $w_0 = 3^{-2k'}/k$ concludes the proof. $\qquad\square$

### D.3. Omitted Details from Section 4.2.1

We restate and prove the following corollary of Lemma 4.4.

**Corollary 4.5.** *Let $p : \mathbb{R} \to \mathbb{R}$ be a polynomial of the form $p(x) = (x - \mu_1)(x - \mu_2) \cdots (x - \mu_{k'})$ where $\mu_1, \dots, \mu_{k'} \in \mathbb{R}$ are arbitrary parameters. Define $I = [0.9\sqrt{2t}, 1.1\sqrt{2t}]$. For all $t \geq 1$ we have $\exp\left(\frac{1}{|I|}\int_{x \in I} \ln|p(x)|\mathrm{d}x\right) \geq \frac{\|p\|_2}{2^{O(k')}}$.*

*Proof.* We can multiply both sides of the conclusion of Lemma 4.4 (Equation (5)) with the Gaussian density $e^{-y^2/2}/\sqrt{2\pi}$ and then integrate both sides. This yields

$$\int_{-\sqrt{t}}^{\sqrt{t}} \frac{1}{\sqrt{2\pi}} e^{-y^2/2} \exp\left(\frac{1}{|I|}\int_{x \in I} \ln|p(x)|\mathrm{d}x\right)\mathrm{d}y \geq \int_{-\sqrt{t}}^{\sqrt{t}} \frac{|p(y)|}{2^{O(k')}} \frac{1}{\sqrt{2\pi}} e^{-y^2/2}\mathrm{d}y \ .$$

The left hand side is $\Theta\left(\exp\left((1/|I|)\int_{x \in I} \ln|p(x)|\mathrm{d}x\right)\right)$. The right hand side is

$$\frac{1}{\sqrt{2\pi}}\int_{-\sqrt{t}}^{\sqrt{t}} e^{-y^2/2}\frac{|p(y)|}{2^{O(k')}}\mathrm{d}y = \left(\mathop{\mathbf{E}}_{y \sim \mathcal{N}(0,1)}[|p(y)|] - \mathop{\mathbf{E}}_{y \sim \mathcal{N}(0,1)}[|p(y)|\mathbb{1}(|y| > \sqrt{t})]\right)2^{-O(k')}$$

$$\geq \left(\mathop{\mathbf{E}}_{y \sim \mathcal{N}(0,1)}[|p(y)|] - \|p\|_2\sqrt{\mathop{\mathbf{Pr}}_{y \sim \mathcal{N}(0,1)}[|y| > t]}\right)2^{-O(k')} \qquad \text{(using the Cauchy–Schwarz inequality)}$$

$$\geq \left(\mathop{\mathbf{E}}_{y \sim \mathcal{N}(0,1)}[|p(y)|] - \|p\|_2 e^{-t^2/2}\right)2^{-O(k')}$$

$$\geq \left(\|p\|_2 - \|p\|_2 e^{-t^2/2}\right)2^{-O(k')} \qquad\qquad\qquad \text{(using Fact 2.4)}$$

$$= \|p\|_2/2^{O(k')} \ . \qquad\qquad\qquad\qquad\qquad\qquad \text{(using } t \geq 1\text{)}$$

$$\square$$

We now restate Lemma 4.4 and provide the complete proof which includes the details that were missing from Section 4.2.1.

**Corollary 4.5.** *Let $p : \mathbb{R} \to \mathbb{R}$ be a polynomial of the form $p(x) = (x - \mu_1)(x - \mu_2) \cdots (x - \mu_{k'})$ where $\mu_1, \dots, \mu_{k'} \in \mathbb{R}$ are arbitrary parameters. Define $I = [0.9\sqrt{2t}, 1.1\sqrt{2t}]$. For all $t \geq 1$ we have $\exp\left(\frac{1}{|I|}\int_{x \in I} \ln|p(x)|\mathrm{d}x\right) \geq \frac{\|p\|_2}{2^{O(k')}}$.*

*Proof.* Fix an arbitrary $y \in \mathbb{R}$ with $|y| \leq \sqrt{t}$. First, note that by the property of logarithms and sums, we can write the left hand side as

$$\exp\left(\sum_{i=1}^{k'} \frac{1}{|I|}\int_{x \in I} \ln|x - \mu_i|\mathrm{d}x\right) \ .$$

In order to show Equation (5), it suffices to work with each term and show the following for each $i \in [k']$:

$$\frac{1}{|I|} \int_{x \in I} \ln|x - \mu_i| \geq \ln|y - \mu_i| - O(1) .$$

Equivalently, it suffices to show that Equation (5) holds for every linear polynomial of the form $p(x) = x - a$. Therefore, the goal for the rest of this proof is to show that

$$\exp\left(\frac{1}{|I|} \int_{x \in I} \ln|x - a| \mathrm{d}x\right) \geq |y - a|/O(1) , \tag{20}$$

holds for every $a \in \mathbb{R}$ and $y \in \mathbb{R}$ with $|y| \leq \sqrt{t}$. We will examine two cases.

**Case 1** The first case is when the root $a$ of the polynomial is outside the interval $I$. In this case, we can show that $|x - a|/|y - a| = \Theta(1)$, which implies $\ln|x - a| \geq \ln|y - a| - O(1)$, and the desired conclusion (Equation (20)) follows by integrating both sides and applying the $\exp(\cdot)$ function.

To show the earlier claim that $|x - a|/|y - a| = \Theta(1)$, we can consider the following sub-cases:

1. Case $a \geq 1.1\sqrt{2t}$ (i.e., $a$ is to the right of $I$): Suppose $a = 1.1\sqrt{2t} + u$ for some non-negative $u$. Then, $a - x = (1.1\sqrt{2t} - x) + u = \Theta(\sqrt{t}) + u$ and $a - y = (1.1\sqrt{2t} - y) + u = \Theta(\sqrt{t}) + u$. Therefore, for any $u \geq 0$, the ratio $|x - a|/|y - a| = (\Theta(\sqrt{t}) + u)/(\Theta(\sqrt{t}) + u) = \Theta(1)$.

2. The cases $a < -\sqrt{t}$ and $a \in [\sqrt{t}, 0.9\sqrt{2t}]$ can be shown in a similar manner.

**Case 2** The complementary case is when the root $a$ of the polynomial $p$ belongs in the interval $I$. In that case,

$$\frac{1}{|I|} \int_{x \in I} \ln|x - a| \mathrm{d}x = \frac{1}{|I|} \int_a^{1.1\sqrt{2t}} \ln(x - a) \mathrm{d}x + \frac{1}{|I|} \int_{0.9\sqrt{2t}}^a \ln(a - x) \mathrm{d}x .$$

Define $A := \frac{1}{0.2\sqrt{2t}} \int_a^{1.1\sqrt{2t}} \ln(x - a) \mathrm{d}x$ and $B := \frac{1}{0.2\sqrt{2t}} \int_{0.9\sqrt{2t}}^a \ln(a - x) \mathrm{d}x$. We will work with each integral separately. For $A$, we have the following (after a change of variable in the integral):

$$A = \frac{1}{0.2\sqrt{2t}} \int_0^{1.1\sqrt{2t} - a} \ln z \, \mathrm{d}z = \frac{1}{0.2\sqrt{2t}} [-z + z \ln z]_{z=0}^{z=1.1\sqrt{2t} - a}$$

$$= -\left(5.5 - \frac{a}{0.2\sqrt{2t}}\right) + \left(5.5 - \frac{a}{0.2\sqrt{2t}}\right) \ln\left(1.1\sqrt{2t} - a\right) .$$

Recalling that we have assumed $a \in [0.9\sqrt{2t}, 1.1\sqrt{2t}]$, we can rewrite the above as $A = -C_1 + C_1 \ln(1.1\sqrt{2t} - a)$, where $C_1 = 5.5 - \frac{a}{0.2\sqrt{2t}} \in [0, 1]$.

We now work with the integral defined as $A$ previously in a similar way:

$$B = \frac{1}{0.2\sqrt{2t}} \int_0^{a - 0.9\sqrt{2t}} \ln z \, \mathrm{d}z = \frac{1}{0.2\sqrt{2t}} [-z + z \ln z]_{z=0}^{z=a - 0.9\sqrt{2t}}$$

$$= -\left(\frac{a}{0.2\sqrt{2t}} - 4.5\right) + \left(\frac{a}{0.2\sqrt{2t}} - 4.5\right) \ln(a - 0.9\sqrt{2t}) .$$

Taking into consideration that $a \in [0.9\sqrt{2t}, 1.1\sqrt{2t}]$ the above can be written as $B = -C_2 + C_2 \ln(a - 0.9\sqrt{2t})$, where $C_2 = \frac{a}{0.2\sqrt{2t}} - 4.5 \in [0, 1]$.

Combining the bounds for $A$ and $B$ together with the definitions $C_1 = 5.5 - \frac{a}{0.2\sqrt{2t}}$ and $C_2 = \frac{a}{0.2\sqrt{2t}} - 4.5$, we obtain $\exp(A + B) = \exp(f(a) - 1)$, where $f(a)$ is the function

$$f(a) := \left(5.5 - \frac{a}{0.2\sqrt{2t}}\right) \ln(1.1\sqrt{2t} - a) + \left(\frac{a}{0.2\sqrt{2t}} - 4.5\right) \ln(a - 0.9\sqrt{2t}) ,$$

We can verify through derivative analysis that the minimum is achieved at the midpoint of $I$, i.e., for $a = \sqrt{2t}$:

$$f'(a) = \frac{5}{\sqrt{2t}} \left( \ln(10a - 9\sqrt{2t}) - \ln(11\sqrt{2t} - 10a) \right) .$$

It is easy to see that $f'(\sqrt{2t}) = 0$. Furthermore, the second derivative is $f''(a) = 1/(t/50 - (a - \sqrt{2t})^2)$, which is non-negative for all $a \in I$. Thus, the only minimizer in $I$ is $a = \sqrt{2t}$. For that point, $\exp(A + B)$ becomes:

$$\exp(A + B) \geq \exp\left( \frac{\ln(t/50)}{2} - 1 \right) \geq \frac{\sqrt{t}}{20} \geq \frac{|y - a|}{52},$$

where we used $|y - a| \leq |y| + |a| \leq \sqrt{t} + 1.1\sqrt{2t} < 2.6\sqrt{t}.$ $\qquad\square$

