# OpenReview forum: "On Learning Parallel Pancakes with Mostly Uniform Weights"
_ICML.cc/2025/Conference — ICML 2025 spotlightposter_

### Official Review · Reviewer_Y4N2 · 2025-03-09

**Overall Recommendation:** 4

**Summary:**

This paper is concerned with learning mixtures of $k$ Guassians, when given i.i.d samples from the mixture. When the mixture weights and covariances of the individual components are unknown and arbitrary, the best-known algorithm from the literature (due to Bakshi et al. 2022) has sample complexity $d^{O(k)}$. A lower bound instance due to Diakonikolas et al. 2017, known as the "parallel pancakes" mixture, comprises of a family of mixtures of $k$ Gaussians having identical covariances. For this family, any SQ based algorithm necessarily requires sample complexity $d^{\Omega(k)}$. However, the weights of the mixture in the family of the lower bound instance end up having to be as small as $2^{-\Omega(k)}$. A natural question here is: can the lower bound be circumvented if the mixture weights are constrained to be at least $1/poly(k)$? Recent results by Anderson et al. 2024, Buhai and Steurer 2023 show that the answer is yes. In particular, they show an algorithm that correctly clusters most points drawn from a $k$-mixture of Gaussians using only $d^{\log(1/w_{min})}$ many samples, where $w_{min}$ is the smallest weight in the mixture.

The first main result in this paper shows that this sample complexity bound is essentially optimal. Theorem 1.3 constructs a family with uniform mixture weights, such that any SQ algorithm must necessarily have sample complexity $d^{\Omega(\log(k))}$.

The algorithms due to Anderson et al. 2024, Buhai and Steurer 2023 have sample complexity $d^k$, even if only a single mixture weight is $2^{-k}$. One can ask if this can be improved. Specifically, consider the setting where $k'$ out of the $k$ mixture weights are allowed to be arbitrary, but the rest are the same. The second main result in the paper (Theorem 1.4) shows that for the task of distinguishing the standard Gaussian from a mixture of this form, there is an algorithm that has sample complexity that scales as $1/{w_{min}}$, which is a better sample complexity that of Anderson et al. 2024, Buhai and Steurer 2023. Note however that the upper bound is only for the testing problem, and not for the learning problem.

For the first result, the authors use a previous construction by Kane 2015, and combine it with a construction from Diakonikolas et al. 2017, to obtain a one-dimensional discrete distribution over $2^{O(t)}$ elements that matches $t$ moments of the standard Gaussian. For the second result, the authors use that their construction for the previous part is essentially optimal. Namely, the uniform distribution on any $k$ points cannot match more that $O(\log k)$ moments of the standard Gaussian. This can be extended to arguing that distributions that are arbitrary on $k'$ out of the $k$ points, but uniform on the rest of $k-k'$ points, cannot match more than $O(\log(k)+k')$ moments of the Gaussian. This fact allows distinguishing between a $k$-mixture from the standard high-dimensional gaussian using higher order tensors that reveal the differences in the higher order moments.

## update after rebuttal
I thank the authors for the clarifications. It would be useful to include them appropriately in the revision of the paper. I maintain my evaluation of the paper, and my score.

**Claims And Evidence:**

The claims and evidence appear convincing to me.

**Essential References Not Discussed:**

NA

**Experimental Designs Or Analyses:**

NA

**Methods And Evaluation Criteria:**

NA

**Other Comments Or Suggestions:**

Please see questions below.

**Other Strengths And Weaknesses:**

The paper is generally quite well-written. Personally, I find that the results in the paper add to the gaps in understanding the sample complexity of learning mixtures of Gaussians, a classical problem in statistics. In particular, the conclusion that the recent postive results due to Anderson et al 2024, Buhai and Steurer 2023 are essentially optimal, is important and satisfying. Furthermore, the additional positive result for the slightly weaker testing problem also indicates algorithms with better dependence on the minimum cluster weight may be attainable for the learning problem. Overall, I find the conclusions in the paper to be important, and they tangibly further our understanding of the gaussian mixture model problem.

**Questions For Authors:**

1) Am I correct in understanding that the general sample complexity upper bound of $d^{O(k)}$ in Bakshi et al. 2022 applies to arbitrary Gaussian mixtures (arbitrary weights, arbitrary means, arbitrary covariances), whereas the lower bound of Diakonikolas et al. 2017 has an additional special property that it is a family of mixtures where for each mixture, the covariances of the components are identical (although means are different, and weights can be exponentially small)?
2) The results of Anderson et al. 2024, Buhai and Steurer 2023 seem to be about clustering points drawn from a mixture model. Could you comment on the differences between this task, and on the task of learning the mixture (i.e., either estimating the parameters/learning a distribution close in TV)?
3) Could you comment on your thoughts on how the positive result for testing (your second result) could extend to a positive learning result? Presently, while the sample complexity of your testing algorithm is better, it is only for a weaker problem. In particular, what are the primary difficulties in employing standard conversions of testing algorithms to learning algorithms?
4) In terms of technical novelty: it seems that beyond the conclusions, the primary technical novelty in the paper is the connection between the construction of Kane 2015, and the past result of Diakonikolas et al. 2017 which shows that one can approximate $t$ moments of the standard Gaussian using a distribution having support only on an interval of size $O(\sqrt{t})$. Has this combination of the Kane 2015 construction and Lemma 3.3 from Diakonikolas et al 2017 been used in the literature in the past?

**Relation To Broader Scientific Literature:**

The problem of learning mixtures of Gaussians is a fundamental problem in statistics that goes back to the days of Karl Pearson from the 1890s, with applications to a variety of sciences. Pinning down the computational and statistical complexity of this problem in different regimes is central to the theory of this problem.

**Theoretical Claims:**

I only glanced over the proofs, and did not verify calculations line-by-line. They appear correct to me, and the progression in the overall analysis checks out to me.

---

> ### Author Rebuttal · Authors · 2025-04-01
>
> We thank the reviewer for their time and positive assessment of our work. We respond to the individual questions below:
>
> (**Difference between Bakshi et al. 2022 and Diakonikolas et al. 2017**) Yes, the reviewer is correct that the algorithm from Bakshi et al. 2022 applies to all GMMs, while in the lower bound of Diakonikolas et al. 2017, the mixtures share a common covariance matrix and potentially different means and weights. This special structure *strengthens* the lower bound, because even the simpler case of common covariance is as hard to learn. We also achieve the common covariance structure in our lower bound, and most importantly, we get a lower bound with *uniform* weights as opposed to the exponentially small weights in Diakonikolas et al. 2017. If one wants to further explore the nuances between common and different covariances, we would like to point out that it is an open and interesting problem to refine the computational landscape based on whether the covariances are the same or not. For example, one open problem that we mention in the paper is whether a learning algorithm with complexity $d^{O(\log(1/w_{\min}))}$ exists that works for arbitrary covariances, which would improve over the $d^{O(k)}$ algorithm by Bakshi et al.
>
> (**Comparison of different kinds of learning guarantees**) The main difference between these guarantees is that for clustering or parameter estimation, one must assume that the mixture components are statistically separated (see the pairwise mean separation assumption in Buhai and Steurer 2023); otherwise, the clustering or parameter estimation goal is information-theoretically impossible. In contrast, if the goal is simply to output a mixture that is close in total variation (TV) distance to the mixture that generated the samples (as in Bakshi et al. 2022), then the separation assumption is not necessary. We emphasize that both the Diakonikolas et al. 2017 lower bound construction and our new construction yield GMMs whose components are indeed pairwise separated, thus they imply lower bounds for all of clustering, parameter estimation and learning-in-TV-distance.
>
> (**Testing -> Learning?**) There are multiple ways to define a learning version of our problem, which we comment on below:
> Learning a parallel pancake distribution in the form defined in Problem 1.1, under the weight restriction assumption of Theorem 1.4 (with $k’$ arbitrary weights and $k-k’$ uniform weights):
>  * First, it should be possible, though not immediately straightforward, to learn the hidden direction $v$ of the parallel pancake distribution (the direction along which the distribution is non-Gaussian). The reasoning behind this is that, since the difference between the population moment tensor of the standard Gaussian and the one for the pancake distribution is the tensor power of the unknown vector $v$, a more sophisticated argument—such as performing tensor SVD to estimate the top eigenvector (in contrast to the simpler argument of our testing algorithm, which just estimates the norm of the moment difference)—might work for learning $v$. Once $v$ is learned, one could attempt to project the samples into direction $v$ to learn the non-Gaussian distribution $A$. There are multiple levels of approximation here: approximating the moments from samples, relating the top eigenvector of the moment tensors to the one from the population version, and bounding the learning error of $A$ along the learned direction. Thus, although the result seems plausible, we have not yet worked out how this propagation of errors can be analyzed.
> * Learning an unknown $k$-mixture of GMMs where k’ weights are arbitrary and $k-k’$ are uniform: This is a much more general problem, thus our testing result provides very limited insight for this problem. As mentioned, there are interesting open problems in this direction.
>
> (**Has the combination of Kane 2015, and Diakonikolas et al. 2017 been used before?**) To the best of our knowledge, the combination of Kane 2015 and Diakonikolas et al. 2017 has not been used in prior work. One key advantage of this approach is that it enables us to establish the lower bound for exactly uniform mixtures, rather than for mixtures with weights that are only polynomially related or differ by a constant factor. Given that we are presenting the first SQ lower bound for equal-weight Gaussian mixtures, it is unlikely that other works would have employed the results and techniques of Kane 2015.

---

### Official Review · Reviewer_4JWr · 2025-03-12

**Overall Recommendation:** 4

**Summary:**

The paper studies a hypothesis testing problem where the main task is to distinguish (with as few as possible samples) between a standard Gaussian $N(0,I_d)$, and a "parallel pancakes" distribution. This distribution is characterized by k discrete mean points along an unknown line in d dimensions. Now the distribution orthogonal to the line is standard Gaussian, but along the line, the variance is squeezed by a factor $1-\delta$.

The main results are:
1. a $d^{\Omega(\log(k))}$ lower bound against distinguishing a standard normal from a pancake mixture where all mixture weights are equal (resp $> 1/poly(k)$). This matches under the same weight assumptions an upper bound of Anderson 2024, $d^{O(\log(1/w_min))}$.

2. When even one $w$ is not bounded below, the upper bound $d^{O(\log(1/w_{min}))}$ becomes $2^{O(k)}$. As a consequent next step the authors provide a testing algorithm with a dominating factor of $(kd)^{O(\log(k) + k')}$ where $k'$ weights are unbounded and $k-k'$ are bounded below by $1/poly(k)$. The minimum weight still plays a role but is only linear in $1/w_min$, so must be super tiny to dominate.

The analyses are highly technical and are based on the observations that there are discrete distributions that share many but not too many smaller moments with the standard Gaussian. The lower bound comes from the fact that the pancakes with the support set as means and spread along a random directional vector are hard to distinguish from the standard normal in d dimensions. And the upper bounds are shown by showing that only a relatively small number of moments are so close that they are indistinguishable. So the algorithm can check all $i$-th moment tensors up to $i \in O(\log(k) + k')$ and must then find at least one larger deviation.

### update after rebuttal:
The authors have addressed my points satisfactorily. Remaining issues are easily resolvable typos. I will therefore retain my initial score "4: Accept".

**Claims And Evidence:**

All claims are rigorously proven.

**Essential References Not Discussed:**

None that I know of.

**Experimental Designs Or Analyses:**

N/A purely theoretical paper

**Methods And Evaluation Criteria:**

N/A purely theoretical paper

**Other Comments Or Suggestions:**

There are a few lines that are unclear to me, possibly typos:
- p6: "A cannot match more than O(log k + k') moments with the standrad Gaussian" - it would be more clear to say that it refers to the *smallest* O(...) moments. Or os there some monotonicity property that I am not aware of (such as if m-th moment matches then also all m'-th moments for m'<m match)?
- p6: \epsilon := \lambda/d^m = (d/\delta)^{(C-1)m}: by the previous bound on lambda, shouldn't the final expression be ((\delta/2)^{C}/d)^m, it seems to be stated inversely?
- line 6 of algo 1 (also algo in the apendix): (i_1 ... j_i), should be (j_1...j_i) or even (1..i) ?
- definition of p(x) is sometimes prod (x-mu_i) (p6) and sometimes prod (x-mu_i)^2 (p7). maybe better to stick to the former and then work with p or p^2 as required.
- p7 l.356: it should be g(x)=f(x), no? f^2 comes only in the next step.

**Other Strengths And Weaknesses:**

The paper is super well written. The high level description of even very technical parts can be followed without prior knowledge in the field.

**Questions For Authors:**

I have no additional questions

**Relation To Broader Scientific Literature:**

Testing of Gaussians and mixtures thereof seems to be an important and active field.

**Theoretical Claims:**

All theoretical claims seem correct.

---

> ### Author Rebuttal · Authors · 2025-04-01
>
> Thank you to the reviewer for the positive assessment of our work and their thorough reading. When we refer to matching the $m$ moments, we always mean the first $m$ moments. We will ensure to clarify this whenever it is not currently explicit. The other points raised are typos, and we will fix them in the final version.

---

### Official Review · Reviewer_Disr · 2025-03-13

**Overall Recommendation:** 4

**Summary:**

This paper studies the problem of learning mixtures of Gaussians where each component in the mixture has a shared covariance.

First, an SQ lower bound is proved, matching a recent positive algorithmic result and indicating that it likely cannot be improved. It is shown that even in the case when the mixture is of $k$ equally weighted Gaussians, the problem of distinguishing the $k$-GMM from a spherical Gaussian has SQ complexity $d^{\Omega(\log(k))}$. This indicates that a recent result showing an algorithm with a $d^{O(\log(1/w_{min}))}$ is likely optimal when $w_{min} = \Omega(1/\mathrm{poly}(k)))$.

Next, this paper works on understanding the optimal complexity dependence on $w_{min}$. It is shown that for instances that are close to uniformly-weighted mixtures (but with a small number of arbitrarily-weighted components), the complexity in $k$ and $w_{min}$ can be somewhat decoupled, and solved with $d^{O(\log(k))}$ operations/samples instead of $d^{O(\log(1/w_{min}))}$ complexity.

**Claims And Evidence:**

Yes, the proofs are clear and I was easily able to follow the general strategy despite this being a very technical work.

**Essential References Not Discussed:**

I am not aware of any

**Experimental Designs Or Analyses:**

N/A

**Methods And Evaluation Criteria:**

Yes, proofs make sense

**Other Comments Or Suggestions:**

* Typo in Fact 3.2, the infimum should be over V_t \setminus \{0\} instead of over V \setminus \{0\}.

**Other Strengths And Weaknesses:**

N/A

**Questions For Authors:**

N/A

**Relation To Broader Scientific Literature:**

There is a large literature on learning Gaussian Mixture Models. This paper represents new fundamental advances in our understanding of the complexity of this problem, which is not yet fully settled. I think that this paper will be well appreciated within its literature.

**Theoretical Claims:**

I checked the proofs in the main text, and they seemed correct

---

> ### Author Rebuttal · Authors · 2025-04-01
>
> We thank the reviewer for their effort and their positive assessment of our work.

---

### Official Review · Reviewer_E3iV · 2025-03-18

**Overall Recommendation:** 4

**Summary:**

This paper studies the hypothesis-testing problem of parallel Gaussian pancakes, specifically under structural assumptions on the component weights. The goal is to distinguish between the standard gaussian and the k gaussian pancakes with collinear centers and common covariance. For learning the general mixture of k gaussians, the best known algorithm have sample complexity $d^{O(k)}$, and in the statistical query model, this problem requires sample complexity $d^{\Omega(k)}$. Recent work considers the setting where the minimum weight of component is $w_{\min}$ and components have common covariance. In this case, they provide $d^{O(\log(1/w_{\min}) )}$ time algorithm to learn the GMM.

In this paper, they first provide a SQ lower bound which shows that even when all component weights are uniform, distinguishing between such a mixture and the standard Gaussian requires $d^{\Omega(\log k)}$ complexity. This implies that the algorithm above is essentially best possible in this case.
Then they provide an algorithm for the hypothesis testing problem when most of the weights are uniform but a small fraction can be arbitrary. Their algorithm has the complexity $(kd)^{O(k'+\log k)} + \log k/w_{\min}$ where $k'$ components have arbitrary weights and the minimum weight is $w_{\min}$. Their algorithm is more efficient than the previous algorithm even if there is one component has an exponentially small weight, like $2^{-k}$. Their results refine existing complexity bounds and offer new insights into the role of weight distributions in learning Gaussian mixtures.

**Claims And Evidence:**

Yes, the claims are all supported by rigorous theoretical analysis.

**Essential References Not Discussed:**

No, the paper cites the related literature thoroughly.

**Experimental Designs Or Analyses:**

N/A

**Methods And Evaluation Criteria:**

Yes, the lower bound and algorithm makes sense and uses various techniques from statistical query complexity, moment-matching analysis, and probability theory.

**Other Comments Or Suggestions:**

Minors:

- Fact 3.2, Line 236, $p \in V_t \setminus$ {0}.
- Lemma 3.5, $p\neq 0$ in the statement, and Line 260, $|x| = O(\sqrt{t})$.
- Line 280, $\epsilon = (\delta / d)^{Cm}$?
- Line 356, g(x) = f(x) ?
- Line 409, $O(\sqrt{t})$ ?
- Appendix D.3, the second Corollary 4.5 should be the restatement of Lemma 4.4.

**Other Strengths And Weaknesses:**

Strengths: The paper provides a solid theoretical contribution by tightening known complexity bounds and introducing novel proof techniques. They also provide a new algorithm that can achieve better complexity for the testing problem of Gaussian pancakes when the weight distribution is structured.

**Questions For Authors:**

1. Given the analysis of the algorithm, is it possible to argue about the lower bound in the case? Is this $O(\log k/ w_{\min})$ term necessary for this problem in the worst case? It seems this is only used to check whether the components are more than $O(\sqrt{d})$ far away from the origin?

**Relation To Broader Scientific Literature:**

The paper builds on prior work in Gaussian mixture model learning, particularly results on statistical query hardness and moment-based learning methods. It extends previous findings by refining complexity bounds and considering more structured weight distributions. The techniques used in the paper can potentially have a broader impact on the learning theory and statistics.

**Theoretical Claims:**

Yes, I checked the correctness of the proofs for theorem 1.3 and 1.4.

---

> ### Author Rebuttal · Authors · 2025-04-01
>
> We thank the reviewer for their effort and their positive assessment of our work. We will fix the typos pointed out in the final version. We respond to their question below:
>
> (**Is the $\log(k)/w_{\min}$ term in the sample complexity necessary?**) This term corresponds to (roughly) the number of samples required to observe at least one sample from each Gaussian component, which is necessary for distinguishing the hypotheses. Suppose that the parameter $\delta$ is sufficiently small such that distinguishing between $N(0, I)$ and $N(0, I - \delta vv^\top)$ requires more than $\log(k)/w_{\min}$ samples. If the parallel pancakes distribution has all but the $i$-th component centered at the origin, then the algorithm cannot make the correct prediction unless it sees a sample from that special component. This shows that $1/w_{i}$ samples are necessary. Since the algorithm does not know which component is the special $i$-th one, our analysis applies a union bound over all components and uses the bound $w_i > w_{\min}$ to arrive at the $\log(k)/w_{\min}$ term in the sample complexity. While there might be some room for improving the $\log(k)$ factor with a potentially better argument than the naive union bound, the $1/w_{\min}$ term is essentially required by the initial argument.

---

### Decision · Program_Chairs · 2025-05-01

**Decision:**

Accept (spotlight poster)

**Comment:**

The paper considers the problem of learning the mixture of k Gaussians with shared unknown covariance and bounded minimum mixture weight. The paper first shows a statistical query lower bound showing that the recent upper bound is tight even when the mixture weights are  uniform. The paper also shows an algorithm for testing whether the given distribution is a mixture of mostly uniform weights or the standard gaussian, with query complexity separating the term required for uniform weight and the term depending on the minimum weight. This is an important evident that more efficient algorithms might be possible for smaller mixture weight lower bounds. All reviewers consider the paper to be a timely contribution to a classical problem and might lead to future improvements in the upper bounds. They also suggest that the paper is well-written.